# Structure and activation mechanism of the Makes caterpillars floppy 1 toxin

Alexander Belyy [1,2], Philipp Heilen[1,2], Philine Hagel[1], Oliver Hofnagel[1] & Stefan Raunser [1]✉

The bacterial Makes caterpillars floppy 1 (Mcf1) toxin promotes apoptosis in insects, leading to loss of body turgor and death. The molecular mechanism underlying Mcf1 intoxication is poorly understood. Here, we present the cryo-EM structure of Mcf1 from *Photorhabdus luminescens*, revealing a seahorse-like shape with a head and tail. While the three head domains contain two effectors, as well as an activator-binding domain (ABD) and an autoprotease, the tail consists of two putative translocation and three putative receptor-binding domains. Rearrangement of the tail moves the C-terminus away from the ABD and allows binding of the host cell ADP-ribosylation factor 3, inducing conformational changes that position the cleavage site closer to the protease. This distinct activation mechanism that is based on a hook-loop interaction results in three autocleavage reactions and the release of two toxic effectors. Unexpectedly, the BH3-like domain containing ABD is not an active effector. Our findings allow us to understand key steps of Mcf1 intoxication at the molecular level.

Many bacteria produce toxins as the major virulence factors during the infection of their hosts. The toxins of insecticidal bacteria have attracted remarkable attention in recent years since they often target specifically insect pests. Therefore, the development and application of insecticidal bacterial toxins have the potential to revolutionize pest management strategies and reduce the harmful effects of chemical insecticides on the environment[1].

To date, the insecticidal toxins from *Bacillus thuringiensis* are the most commercially successful organic pest control agents[2]. However, the emergence of resistance stimulates the search for novel biopesticides[3]. Promising alternatives include highly efficient insecticidal toxins from entomopathogenic bacteria of the *Photorhabdus* genus: Tc toxins, large bacterial protein complexes that translocate cytotoxic enzymes into target cells using a syringe-like mechanism[4,5]; *Photorhabdus* Insect Related (Pra/Prb) Toxins, binary toxins similar to *Bacillus thuringiensis* δ-endotoxins[6]; *Photorhabdus* Virulence Cassettes (PVC), phage-like protein-translocation structures that inject effector proteins into target cells[7]; and "Makes caterpillars floppy 1" toxin (Mcf1), which induce cell apoptosis by a mechanism that is not yet

understood[8]. In addition to potential agricultural applications, the modular nature of these toxins allows them to be converted into medical delivery systems in gene and cancer therapy to target molecules of interest into cells[9,10].

Mcf1 from *P. luminescens* (Mcf1Ac1[11,12] in the following called Mcf1) is a 324 kDa protein, which was discovered when *Manduca sexta* larvae were infected with *Escherichia coli* carrying fragments of the *P. luminescens* genome[8]. At the cellular level, activity of the toxin leads to apoptotic blebbing of membranes and fragmentation of nuclei[13]. Such apoptosis in the midgut cells leads to a rapid loss of body turgor and the characteristic "floppy" phenotype of the intoxicated insect within 12 h and death of the organism at around 24 h after intoxication[8].

Mcf1 shares limited similarity to known proteins in four regions, as revealed by a whole-genome search. First, the N-terminal region contains a sequence of 15 amino acids (residues 911–925) resembling the BH3 domain of pro-apoptotic proteins. Therefore, it was logical to propose that this region induces apoptosis during cell intoxication with Mcf1[12]. Second, the central region of Mcf1 shares similarities with the 50 kDa Mcf-like effector domain of the multifunctional-autoprocessing

[1]Department of Structural Biochemistry, Max Planck Institute of Molecular Physiology, Otto-Hahn-Str. 11, 44227 Dortmund, Germany. [2]These authors contributed equally: Alexander Belyy, Philipp Heilen. ✉e-mail: stefan.raunser@mpi-dortmund.mpg.de

repeats-in-toxin (MARTX) toxins of *Vibrio vulnificus*[14,15]. This domain is a cysteine protease that, upon autoproteolytic activation by host cell ADP-ribosylation factor (Arf) proteins, degrades Rab GTPases and induces cell apoptosis[16]. Third, Mcf1 possesses a highly hydrophobic sequence similar to that in the translocation region of the *Clostridioides difficile* toxins A and B (TcdA, TcdB), suggesting that Mcf1 can undergo a conformational change and penetrate the endosomal membrane, similar to these toxins[17]. Finally, the C-terminal region shows minimal similarity to the putative receptor-binding regions of RTX-like toxins[13] (Supplementary Fig. 3e).

Interestingly, some *P. luminescens* strains also express a Mcf1 homolog, called Mcf2. These two toxins are similar in the central and C-terminal regions but differ strikingly at their N-terminal regions. The N-terminal region of Mcf2 does not contain a BH3-like domain, but probably exerts its activity through the HrmA-like domain of the plant pathogen *Pseudomonas syringae*[18]. However, the structure, mechanism of cellular internalization and molecular mechanism of action of Mcf1 and Mcf2 remain unknown.

In this study, we focus on Mcf1, determine its cryo-EM structure and functionally analyze the toxin. The structure reveals that Mcf1 is divided into an N-terminal head region and a C-terminal tail region. The head contains an N-terminal effector domain (NED), an activator-binding domain (ABD), and a protease effector domain (PED). The tail consists of two putative translocation and three putative receptor-binding domains. After translocation into target cells, Mcf1 interacts with the host cell ADP-ribosylation factor 3 (Arf3) that allosterically

activates the PED using a distinct pulling mechanism. The latter promotes three cleavage reactions and the release of two toxic effectors into the cytoplasm. Taken together, our results provide molecular insights into the architecture of Mcf1 and allow us to understand key steps of its mechanism of action.

## Results

### The architecture of Mcf1

To understand how Mcf1 is built in molecular detail, we heterologously expressed and purified the full-length Mcf1 complex from *Photorhabdus luminescens* and determined its structure using single particle cryo-EM to 3.6 Å resolution (Supplementary Table 1, Supplementary Fig. 1, Supplementary Movie 1, Methods). Since the toxin proved to be inherently flexible, we performed a series of signal subtraction and local refinements of subregions of the complex. The resulting composite reconstruction allowed us to build 92% of the Mcf1 atomic model de novo. An additional dataset of C-terminally truncated Mcf1 (Methods) enabled us to build the remaining N-terminal region, completing the atomic model of Mcf1 encompassing 2,911 of 2,929 amino acids.

The structure reveals that Mcf1 is a monomeric, asymmetric protein. Based on its overall shape, reminiscent of a seahorse, it can be divided into two parts - an N-terminal head and a C-terminal tail region (Fig. 1). Both are composed of several domains. In order to name them and identify their potential function, we compared the sequence, the general architecture and the structure of the domains with those of

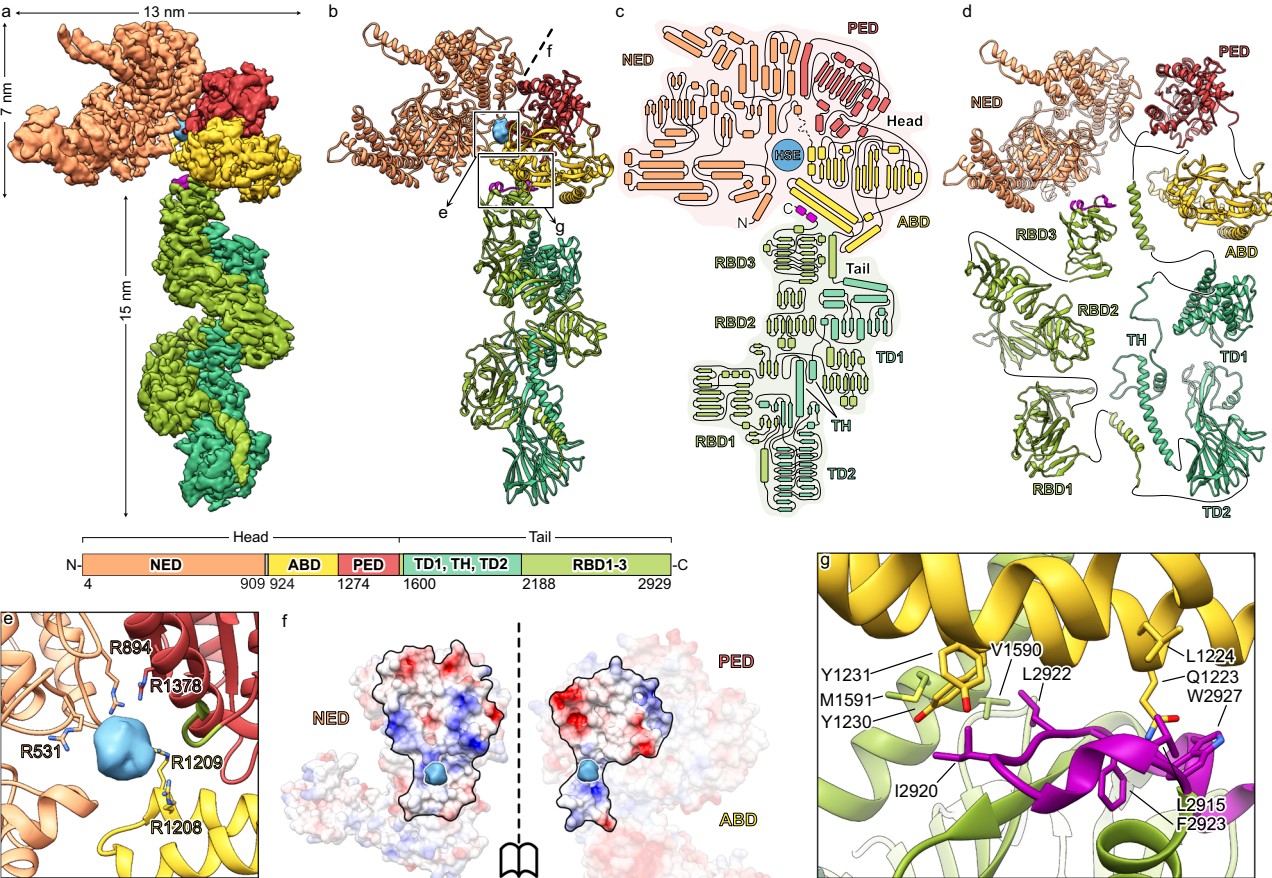

**Fig. 1 | The structure of Mcf1.** Composite cryo-EM density (**a**) and atomic model (**b**) of Mcf1. Schematic representation of the secondary structure (**c**) and domain architecture (**d**). **e** Density and coordination of the head-stabilizing element. **f** Surface at the interface of the three domains of the Mcf1 head (N-terminal effector domain, protease effector domain and activator-binding domain) colored by electrostatic Coulomb potential from −10 kcal mol⁻¹ (red) to +10 kcal mol⁻¹ (blue). **g** Close-up view of the neck region, connecting the head and tail of Mcf1 (regulatory C-terminus - magenta). NED N-terminal effector domain, ABD activator-binding domain, PED protease effector domain, RBD receptor-binding domain, TD translocation domain, TH transmembrane helices, HSE head-stabilizing element.

other proteins (Supplementary Figs. 2, 3). This analysis revealed that Mcf1 is built in a similar fashion to TcdA and TcdB toxins from *C. difficile*[19] (Supplementary Fig. 3a).

The N-terminal domain of TcdA and TcdB comprises an effector domain, namely a glycosyltransferase[20,21]. Although the equivalent domain of Mcf1 has no structural or sequence homology to this domain, we named it N-terminal effector domain (NED). Indeed, a subregion of this domain has a structural homology to bacterial ADP-ribosyltransferases (RhsP2 from *Pseudomonas aeruginosa*[22] and Rhs1 from *Salmonella enterica*[23]) (Supplementary Fig. 2).

The second domain of the Mcf1 head is not found in TcdA or TcdB. It has structural homology to the *Burkholderia* lethal factor 1[24], but one critical catalytic residue is missing (Fig. 3a). Therefore, and due to studies that will be described below, we dubbed this domain activator-binding domain (ABD). The third domain of the Mcf1 head has a high homology to the cysteine protease domain (CPD) of TcdA and TcdB from *C. difficile* (Supplementary Fig. 2). In addition, it is homologous to the Mcf-like domain of the multifunctional-autoprocessing repeats-in-toxin (MARTX) toxin from *Vibrio vulnificus*, which induces cell apoptosis by cleaving C-terminal regions of Rab proteins[16,18]. We therefore named this domain protease effector domain (PED), reflecting its potential dual function.

The NED and the PED have a large interaction interface formed mostly by electrostatic interactions (Fig. 1f). In addition, we noticed a large spherical density of 10 Å diameter in the middle of the head that is coordinated by five arginines from all three head domains (Fig. 1e). Its location at the interface of the three domains and coordination suggests that this is a negatively charged molecule, which stabilizes the head region. We therefore call it head-stabilizing element (HSE).

The elongated tail starts with an α-helical domain and a β-strand-rich domain (Fig. 1c, d), separated by a segment of 99 amino acids, 87 of which are hydrophobic, with high probability to form transmembrane helices (Supplementary Fig. 3b). Indeed, these domains have a striking structural similarity to the translocation apparatus of TcdA and TcdB toxins from *C. difficile* (Supplementary Fig. 2). We therefore termed them translocation domain 1 (TD1), transmembrane helix (TH) and translocation domain 2 (TD2). Interestingly, the putative translocation domains of Mcf1 match the recently derived consensus sequence of the evolutionary conserved translocase of large clostridial toxins (LCTs) even better than the original *C. difficile* toxins[17] (Supplementary Fig. 3c). Since the common translocation apparatus is the hallmark of the LCTs, our structural data clearly identify Mcf1 as a member of this bacterial toxin superfamily.

The tail of Mcf1 ends with three β-strand-rich domains. The latter one as well as translocation domain 2 show structural similarity to the β-roll structures of RTX toxins without sharing their consensus sequence and characteristic $Ca^{2+}$ binding[25]. Since the last domain also shares structural similarity to a region of *C. difficile* toxins that has been connected to receptor binding (Supplementary Fig. 2), we called these domains putative receptor-binding domains 1–3 (RBD1-3) (Fig. 1c, d). Interestingly, these putative receptor-binding domains are arranged such that they fold back onto the tail towards the head, resulting in the C-terminus of Mcf1 to be located within 2 nm distance from its N-terminus at the neck region (Fig. 1c, d). Moreover, the C-terminal region forms multiple hydrophobic interactions with a helix bundle of the ABD and supports the helix that directly connects the head and the tail region of the toxin, thus organizing the neck region (Fig. 1g). This particular fold is likely an important element of the toxin.

## Autoproteolysis of Mcf1
The structure revealed that Mcf1 possesses a complete protease domain. It has a typical α/β hydrolase fold with a correctly positioned catalytic triad comprising a cysteine, histidine and aspartic acid (Fig. 3a) indicating that it likely functions as a protease. To test if this domain is active and responsible for Mcf1 processing as in the case of

other toxins[26,27], we intoxicated Sf9 insect cells with N-terminally or C-terminally Myc-tagged Mcf1 variants and monitored the toxin during intoxication using western blots of cell lysates. Wild-type Mcf1 was cleaved into a ~105 kDa N-terminal and ~150 kDa C-terminal fragment in a time-dependent manner (Fig. 2a), demonstrating that Mcf1 is proteolyzed inside the target cell. Interestingly, the C1379A variant of Mcf1, which impairs the catalytic center of the domain, remains uncleaved, indicating that the PED is indeed responsible for the cleavage of Mcf1 by autoproteolysis.

When mapping the fragment sizes on our structure, we found that the N-terminal 105 kDa fragment lacks the ABD and PED and only corresponds to the NED. The C-terminal 150 kDa fragment matches the tail region (Fig. 2b). Thus, the experiment demonstrates that Mcf1 is cleaved at least at two positions inside the cytoplasm of the target cell. This is clearly different from *C. difficile* TcdA and TcdB that have a single cleavage site[26,27].

## Arf3 triggers Mcf1 autoproteolysis
Although we could demonstrate that Mcf1 is autoproteolytically cleaved in cells, the toxin was stable in solution for long periods of time. This indicates that the protease apparently needs to be activated by factor(s) of the target cell. In the case of TcdA and TcdB, the protease domain directly interacts with the eukaryote-specific molecule inositol hexakisphosphate (IP6) to activate the catalytic center by orienting the catalytic residues[26,28]. However, based on our structural data, Mcf1 does not have an apparent IP6-binding site and its catalytic center does not require reorganization since all three residues of the catalytic triad are in 3 Å proximity. Therefore, the mechanism of Mcf1 activation must be completely different from the one of LCT toxins.

To reveal this mechanism, we aimed to identify the host cell factor(s) that leads to the activation of the protease and subsequent processing of the toxin. The 50 kDa Mcf-like domain of MARTX from *V. vulnificus* binds to eukaryote-specific ADP-ribosylation factors (Arfs) in their GTP-bound state with nanomolar affinity[14,15]. Due to the structural homology, we chose one of these factors, Arf3, which is primarily located on the surface of the host Golgi apparatus and endosomes[29], and tested whether it could stimulate proteolytic cleavage of Mcf1 in vitro. Incubation of N-terminally His-tagged and C-terminally FLAG-tagged Mcf1 with His-tagged Arf3 indeed resulted in the cleavage of Mcf1, indicating that Arf3 acts as an activator of Mcf1 autoproteolysis (Fig. 2c).

In the case of the Mcf-like domain of MARTX, it was shown that Arf binding mechanically moves the cleavage site into the catalytic center and thereby triggers cleavage[15]. Such a mechanism is not possible for Mcf1 since binding of Arf3 to the PED is sterically hindered by the presence of the NED (Supplementary Fig. 4a). This suggests, that Arf3 binding to Mcf1 activates autoproteolysis in a different way.

## Neck region regulates Mcf1 processing
We noticed that the autoprocessing reaction in the presence of Arf3 was much slower in vitro than during cell intoxication (Fig. 2a, c), suggesting that either another eukaryotic cell factor is required for the reaction, or a conformational change in Mcf1 during membrane insertion or translocation through the host cell membrane is necessary to stimulate the PED.

First, to address the requirement for another eukaryotic host cell factor, we developed a simple in vivo system, in which we co-expressed the *arf3* and *mcf1* genes from a single plasmid in *E. coli*. We then took advantage of the Myc-tag at the N-terminus of Mcf1 to monitor its processing. We observed rapid and efficient Arf3-dependent toxin processing, confirming that Arf3 is the only essential eukaryotic factor in this *E. coli* system (Supplementary Fig. 4b). This rules out that another host cell factor is required for efficient Mcf1 autoproteolysis.

We hypothesize that membrane insertion and translocation of Mcf1 during intoxication of insect cells would inevitably result in

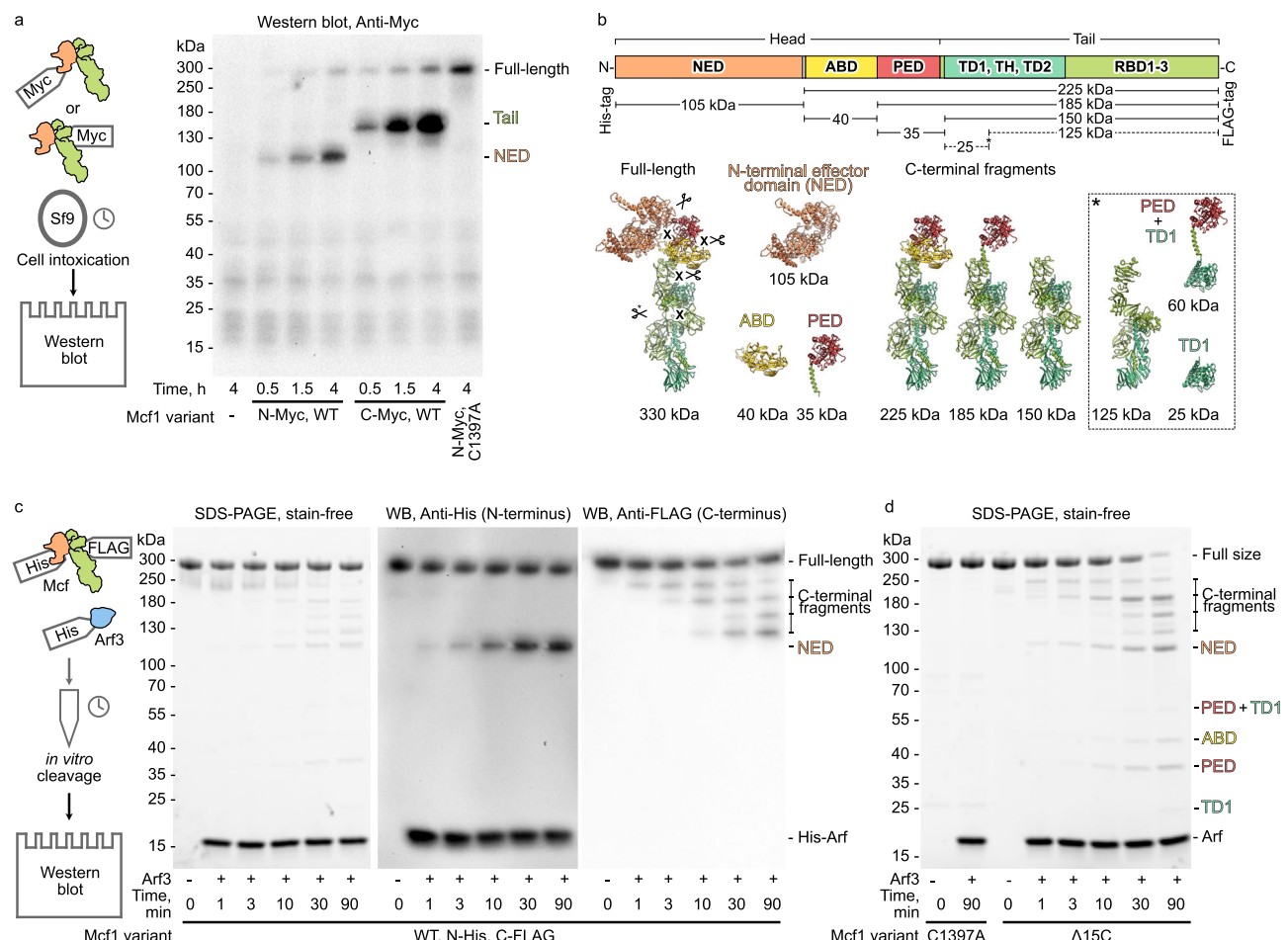

**Fig. 2 | Autoproteolytic cleavage of Mcf1. a** Western blot analysis of Sf9 cells intoxicated with wild-type (WT) or C1397A Mcf1 variants. The proteins were Myc-tagged at the N- (N-Myc) or C-terminus (C-Myc). **b** Mapping of the expected cleavage products on the Mcf1 structure. Approximate molecular weights of the fragments are indicated. **c** SDS-PAGE and western blots (WB) of the Arf3-dependent autoproteolysis of N-terminally His-tagged (N-His) and C-terminally FLAG-tagged (C-FLAG) Mcf1 in vitro. **d** Autoproteolysis of C-terminally truncated (Δ15C) Mcf1 variants in vitro. Source data are provided as a Source Data file. The cleavage experiments and their corresponding western blot analysis were performed twice. Abbreviations: see Fig. 1.

conformational changes at the neck region, where the C-terminus of the protein makes strong hydrophobic interactions with the head of Mcf1 (Fig. 1g). To test the role of the C-terminal region of Mcf1 in the autoprocessing reaction, we purified a Mcf1 mutant devoid of 15 C-terminal amino acids (Δ15C) and incubated it with Arf3 in vitro. Strikingly, the autocleavage reaction proceeded several times faster than with the wild-type toxin and yielded the same fragments (Fig. 2d). This result demonstrates that the C-terminus of Mcf1 is a potential regulatory element that prevents autoprocessing of Mcf1 prior to its translocation into the target cell.

## Mcf1 undergoes three autocleavage reactions

In our cell intoxication and in vitro cleavage experiments, we detected either two or multiple fragments of Mcf1, which is surprising given the fact that TcdA and TcdB from *C. difficile* are only cleaved at one position[30] (Fig. 2a). By combining the lengths of the different fragments and mapping them on our structure, we found that cleavage must occur at three different positions resulting in the separation of the NED, ABD, PED, and the Mcf1 tail (Fig. 2b).

We used mass spectrometry to locate the first cleavage site (Methods) and found that cleavage occurs between lysine 912 and alanine 913 and results in the release of the 105 kDa NED (Supplementary Fig. 4c). To confirm this, we intoxicated cells with the cleavage site-impaired Mcf1 L911A K912A variant (Supplementary Fig. 4d, e) and noticed a shift in the size of the N-terminal fragment from 105 kDa to

145 kDa, corresponding to the NED and ABD. Although mass spectrometry could not clearly determine cleavage site 2 and 3, we could narrow down cleavage site 2 by site-directed mutagenesis to the motif 1271-IQGG-1274 (Supplementary Fig. 4d, e).

Interestingly, in our in vitro assay we noticed one more 125 kDa C-terminal fragment, which did not appear in the cell intoxication assays (Fig. 2c, d). Mapping of the fragment on the structure suggested that it represents the tail region without the putative translocation domain 1. In contrast to the in vitro conditions, where the PED can access this region and excise the putative translocation domain 1, during cell intoxication this domain is likely membrane-bound and therefore inaccessible to the protease. Such a scenario would further corroborate the idea that the conformational change during membrane penetration precedes toxin activation inside the host cells.

## N-terminal effector and protease effector domains are toxic in eukaryotic cells

Intoxication of insect and mammalian cells with Mcf1 leads to apoptotic blebbing of membranes and fragmentation of nuclei[13] (Supplementary Fig. 3d). To find out which of the three fragments that Mcf1 excises from the membrane-embedded tail domain are actually cytotoxic in eukaryotic cells, we expressed these fragments in the *Saccharomyces cerevisiae* model that we previously established to study various bacterial effectors (i.e., TccC3[31], ExoY[32,33] and Iota-A[34]).

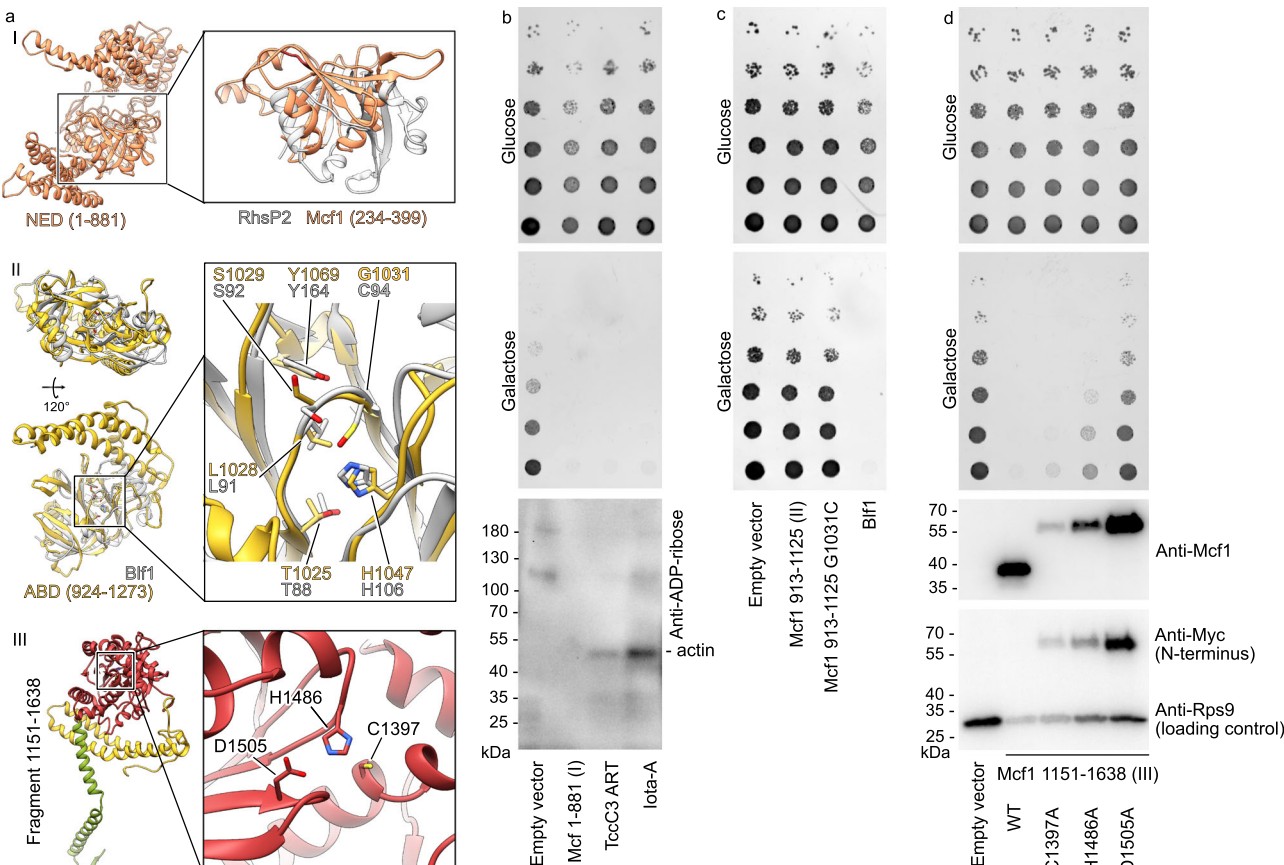

**Fig. 3 | Toxicity of Mcf1 effectors. a** Structure of the three head domains and comparison with structural homologs of two of them[22,24]. **b, c, d** Growth phenotype assay with *S. cerevisiae* expressing Mcf1 fragments under galactose promoter in the experimental conditions with low (glucose) or high (galactose) Mcf1 fragment expression. The protein expression and the level of ADP-ribosylation was analyzed by western blot of cells grown on galactose-containing media with anti-Myc, anti-Mcf1, ADP-ribose binding reagent (anti-ADP-ribose) and anti-ribosomal protein S9 (RPS9) antibodies. Blf1 *Burkholderia* lethal factor 1, NED N-terminal effector domain, ABD activator-binding domain, PED protease effector domain, ART ADP-ribosyltransferase. Source data are provided as a Source Data file. Western blot analysis was performed twice.

Our analysis of yeast viability demonstrated that expression of the 105 kDa NED was highly toxic (Fig. 3b). Interestingly, the NED could not be split or truncated for more than a dozen amino acids without loss of function, revealing that, despite its relatively large size, this domain acts as a single toxic effector (Supplementary Fig. 5a). Despite its structural homology with ADP-ribosyltransferases (Supplementary Fig. 2), the NED showed no detectable ADP-ribosyltransferase activity in yeast or in insect cells (Supplementary Figs. 5b, 3b), as well as no affinity to NAD⁺ when measured by isothermal titration calorimetry (Supplementary Fig. 6), indicating that it must have a different mechanism of action.

The second fragment released by Mcf1 corresponds to the 40 kDa ABD (Fig. 2b, d). Interestingly, this domain also contains a putative BH3-like domain (amino acids 911–925)[12]. Due to the function of BH3-like proteins in apoptosis, it has been speculated that this part of Mcf1 mediates toxicity in cells in a similar fashion. In addition, the ABD is structurally homologous to the *Burkholderia* lethal factor 1 (Blf1), which is a highly potent deamidase[24]. In contrast to the highly toxic Blf1, production of the ABD did not affect yeast growth (Fig. 3c). Yeast growth was also not affected by reintroduction of a catalytic cysteine, which is missing in the ABD but required for the deamidation reaction of Blf1 (Fig. 3a). Therefore, we conclude that this part of Mcf1 does not induce apoptosis and is in general not an active effector, at least in yeast cells. We speculate that this domain lost its enzymatic activity but remained an important structural element of the toxin.

Finally, we expressed a 56 kDa fragment (amino acids 1133–1619), which includes the complete PED, in yeast. Surprisingly, in addition to the fact that the domain dramatically inhibited cell growth (Fig. 3d), the toxic PED was truncated from the N-terminus to 40 kDa (Fig. 3d). We then tested whether the reduced yeast growth was related to the proteolytic activity of the PED by mutating its catalytic residues. Our viability assays demonstrated that mutations in the catalytic center of the PED decreased cell toxicity (Fig. 3d). Therefore, our experiments suggest that the PED, matching its name, plays a dual function: it activates the Mcf1 toxin by proteolytic cleavage, and also directly mediates toxicity in cells. Interestingly, a similar protease domain from the *P. asymbiotica* PaTox toxin was shown to enhance the cytotoxic effects of the toxin, but did not show any toxic effects alone[35]. Therefore, a dual role of protease domains might be a key feature of Mcf-like toxins from *Photorhabdus* that allows to increase their overall potency.

### The mechanism of Mcf1 activation

We have shown that the interaction of Mcf1 with Arf stimulates the autoproteolytic activity of the toxin and the release of three fragments, at least two of which induce cell death. However, the molecular mechanism of this activation remains unclear, since the first cleavage site is more than 20 Å away from the catalytic center of the PED. Thus, for the proteolytic reaction to occur, either the linker with the cleavage site or the PED must be moved (Fig. 4d). To decipher the molecular mechanism of the activation process, we analyzed the Mcf1-Arf3

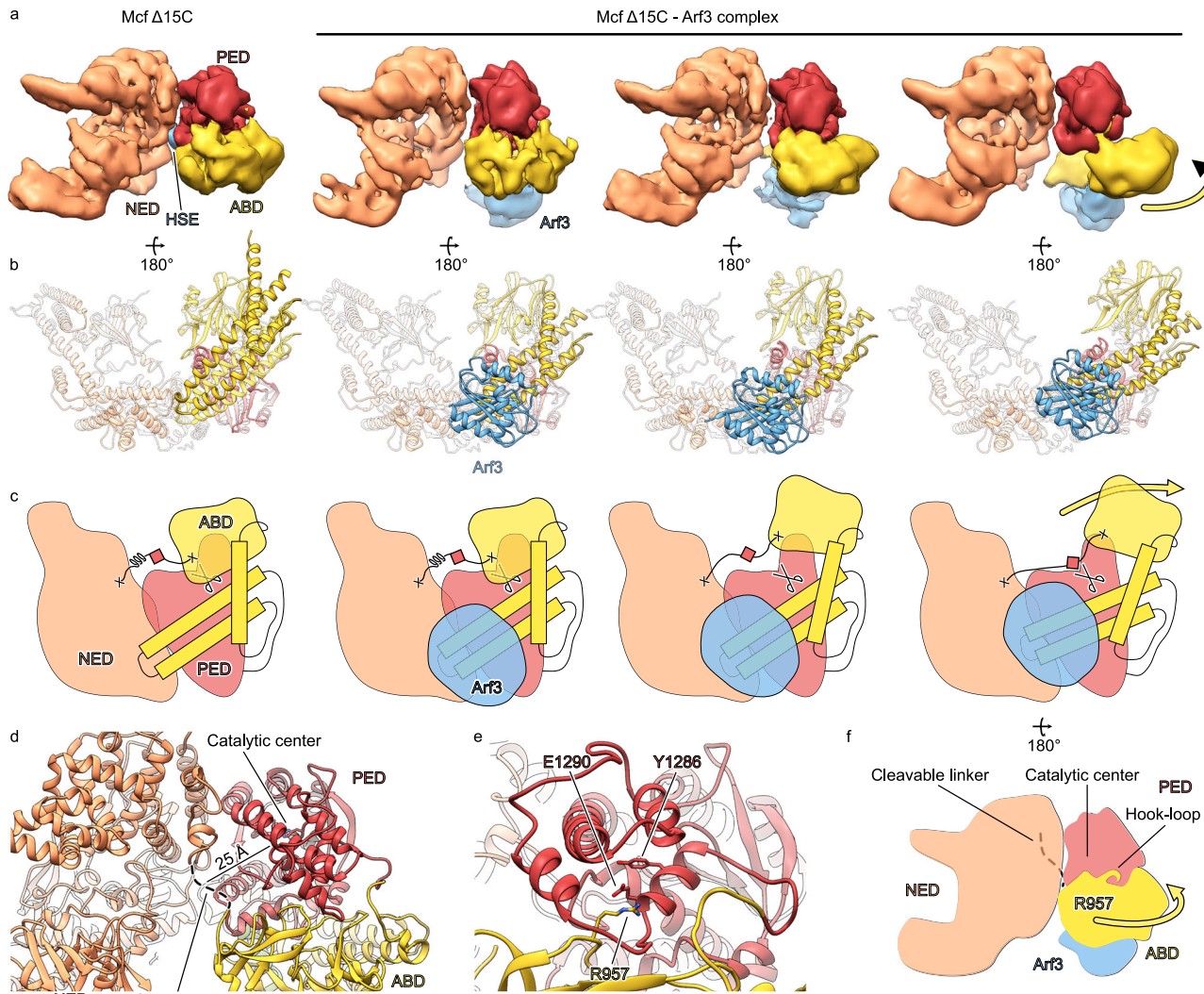

**Fig. 4 | Mechanism of activation of Mcf1.** Different 3D class averages (**a**) and corresponding molecular models (**b**) of the Mcf1$_{C1397A \Delta15C}$ and Mcf1$_{C1397A \Delta15C}$-Arf3 complex datasets. Schematic illustration of the mechanism of Mcf1 activation from the bottom (**c**) and front (**f**). Upon Arf binding, helices of the ABD are strained, pulling the linker towards the active site, while being guided by the hook-loop interaction. **d** The cleavable linker is localized in 25 Å distance from the catalytic center of the protease effector domain (PED). **e** Structure of the hook (R957) – loop interaction of Mcf1. Abbreviations: see Fig. 1.

interaction by cryo-EM and single particle analysis. To obtain a stable complex, we used the C1397A Δ15C variant of Mcf1 to make sure that the protease is inactive and the C-terminus does not interfere with the interaction (see above) and determined structures of the Mcf1 variant in the absence and presence of Arf3 (Supplementary Figs. 8, 9; Supplementary Table 1).

During particle selection and 2D classification, we noticed in both datasets that the Mcf1 head domain was clearly visible, while the tail domain was absent in most 2D classes or blurry in others (Supplementary Figs. 8b, 9b). This is in contrast to the single particle analysis of the wild type (Supplementary Fig. 1b) and suggests, that the removal of the last 15 amino acids destabilizes the neck regions leading to a high flexibility between the head and tail. Therefore, we only concentrated on the heads for the following analysis. During 3D refinement and classification, we noticed that the heads were also very heterogenous and the ABD was missing in most classes of Mcf1$_{C1397A \Delta15C}$. Nevertheless, we managed to determine a cryo-EM structure of the complete head of Mcf1$_{C1397A \Delta15C}$ in absence of Arf3 at 3.5 Å resolution (Supplementary Fig. 8b).

In contrast to the head in absence of Arf3, the ABD was present in the Mcf1$_{C1397A \Delta15C}$-Arf3 map, and surprisingly, we observed an additional density close to the ABD and not to the PED. This suggests that

Arf3 binds to the ABD and thereby stabilizes it. We then determined a cryo-EM structure of the complete head of Mcf1$_{C1397A \Delta15C}$-Arf3 at a resolution of 4 Å that let us unambiguously fit Arf3 in the additional density. The atomic model of the Mcf1-Arf3 interface revealed that Arf3 binds to all three helices of the ABD (Supplementary Fig. 7a), which is the same region where the C-terminus is located in wild-type Mcf1 (Fig. 1g). This observation provides a clear explanation why deletion of the extreme C-terminus enhanced the autoproteolysis rate of Mcf1 (Fig. 2d) and therefore defines the C-terminus as important regulator of Mcf1 autoprocessing.

The comparison of the Mcf1$_{C1397A \Delta15C}$-Arf3 structure with Mcf1$_{C1397A \Delta15C}$, however, showed that the position of the protease domain is not altered and the cleavable linker is more or less in the same position, indicating that Arf3 binding does not induce direct conformational changes. We therefore went back to further analyze our 3D classes and found that Arf binding induces a swinging movement of the ABD away from the NED (Fig. 4a, Supplementary Fig. 9b, Supplementary Movie 2). Interestingly, the density for the NED and the PED was of higher resolution and appeared at higher binarization thresholds, suggesting structural rigidity of these domains. Therefore, the movement of the ABD can lead to displacement of the cleavable linker towards the protease domain where it is proteolytically cleaved.

On a molecular level, we suggest that Arf3 binding strains the helices of the ABD and drives the movement of its N-terminal part (Fig. 4b, c). This movement is possible due to a hook-loop interaction between the ABD and the PED, organized by arginine 957 of the ABD and the loop 1291-1298 of the PED (Fig. 4e and f). We propose that this structural feature allows the ABD to swing away from the N-terminal effector, and to pull the cleavable linker toward the catalytic center of the PED for proteolysis (Fig. 4c and f).

To test this hypothesis, we first studied the role of the hook-loop interaction between the activator-binding and PED (Fig. 4e). To this end, we engineered two Mcf1 mutants: in the first one we aimed to abolish the interaction between the activator-binding and PEDs by mutating arginine 957 to alanine, and in the second one we introduced a mutation in tyrosine 1286 to make the loop more flexible and increase the freedom of movement of R957 during activation. When we then incubated these Mcf1 variants with Arf3, we observed nearly no autoprocessing of the R957A mutant and instant cleavage of the Y1286S variant (Supplementary Fig. 7b). These results clearly indicate the importance of the hook-loop interaction in coordinating the ABD movement and linker stretch.

Finally, we aimed to test the requirement of pulling the cleavable linker for Mcf1 activation. To this end, we engineered two Mcf1 mutants with one or two amino acid deletions (ΔK907 and ΔD908) upstream of the cleavage site to shorten the linker and another two Mcf1 mutants L1359G and R1360G to destabilize the rigid base that keeps the N-terminal effector and the PED rigid and immobile (Supplementary Fig. 7c). We then incubated these four mutants with Arf3 and found drastically reduced autoproteolytic activity (Supplementary Fig. 7d, e), thus demonstrating that the pulling of the linker is an essential step in toxin autoproteolysis. Altogether, our structural and biochemical data reveal a distinct activation mechanism of a bacterial toxin through allosteric displacement of the cleavage site toward the catalytic center of the protease induced by Arf (Fig. 4c, f).

## Discussion

In this study, we present a detailed structural and functional analysis of the Mcf1 toxin from *P. luminescens*, which allows us to confidently assign Mcf1 to the ABCD subfamily of bacterial toxins[30] next to TcdA and TcdB toxins from *C. difficile*, which is a major cause of health care-associated diarrhea and colitis[36].

Mcf1 and LCTs share similarities in their domain architecture (Supplementary Figs. 2, 3a). However, Mcf possesses three putative receptor-binding domains instead of an extended and mobile CROPs domain, an additional ABD and a N-terminal effector with no homology to the glycosyltransferase of TcdA or TcdB[37]. Finally, we demonstrate that the protease domain, which is normally required only for autoprocessing, is a toxic effector of Mcf1 per se, unlike the structurally similar CPD of the PaTox toxin from *P. asymbiotica*, which was

shown to increase the toxic effect of PaTox without being toxic itself[34]. We speculate that the mechanism of toxicity of the PED might be similar to the Mcf-like domain of *V. vulnificus* MARTX, which induces cell apoptosis by cleaving C-terminal regions of Rab proteins[16]. Thus, Mcf1 provides an example of an evolutionary scramble between different toxins of different families and highlights the potential of its modular architecture for the design of customizable delivery systems.

Our structural and functional data together with previous analyses by other groups allow us to propose a molecular mechanism of action of Mcf1 (Fig. 5). After binding of the toxin to a-yet-unknown protein and/or glycan receptors on the surface of the host cell, the toxin is endocytosed[13] similar to TcdA and TcdB[38-40]. Since there are no large charged regions on the toxin that would form contact with the membrane to orient it in a specific way (Supplementary Fig. 10), such an interaction is likely facilitated by the receptor. Acidification of the endosome triggers a conformational change inside the 600 amino acid-long delivery region, which forms a pore and translocates the head region of the toxin inside the target cell as it is the case for other members of the LCTs[17,41]. After translocation, the ABD of Mcf1 has lost its interaction with the C-terminal region, which probably remains on the other side of the membrane. Therefore, the binding site for membrane-bound Arf proteins on the ABD becomes available for interaction. Interestingly, the Mcf-like effector of MARTX from *V. vulnificus* is also activated by Arf binding. However, in that case Arf directly interacts with the protease domain (Supplementary Fig. 4a), leading to a different mechanism of activation[15]. Here, formation of the Mcf1-Arf3 complex stimulates a swinging movement of the ABD that pulls the cleavable linker towards the catalytic site of the PED. After the first cleavage of the NED, the PED catalyzes two more proteolysis reactions resulting in the release of the PED and ABD. While the release of the NED is similar to the glycosyltransferase cleavage from TcdA or TcdB[27,42], the further cleavages are more resembling the processing of the MARTX toxin from *V. vulnificus* that releases the Mcf-like fragment[15,43]. Enzymatic activity of the Mcf1 effectors in the cytoplasm of the target cells impairs essential physiological pathways and ultimately leads to cell death. We believe that our structural and mechanistic insights will help to engineer Mcf1 variants for ecological pest control.

## Methods

### Plasmids, bacteria and yeast strains

The complete list of oligonucleotides, plasmids and strains used in this study can be found in Supplementary Table 2. *E. coli* were cultured in LB medium supplemented with kanamycin or ampicillin. *S. cerevisiae* were cultivated on synthetic defined medium (Yeast nitrogen base, Difco) containing galactose or glucose and supplemented if required with histidine, uracil, tryptophan, adenine, or leucine. Yeast transformation was performed using the lithium-acetate method[44]. Yeast

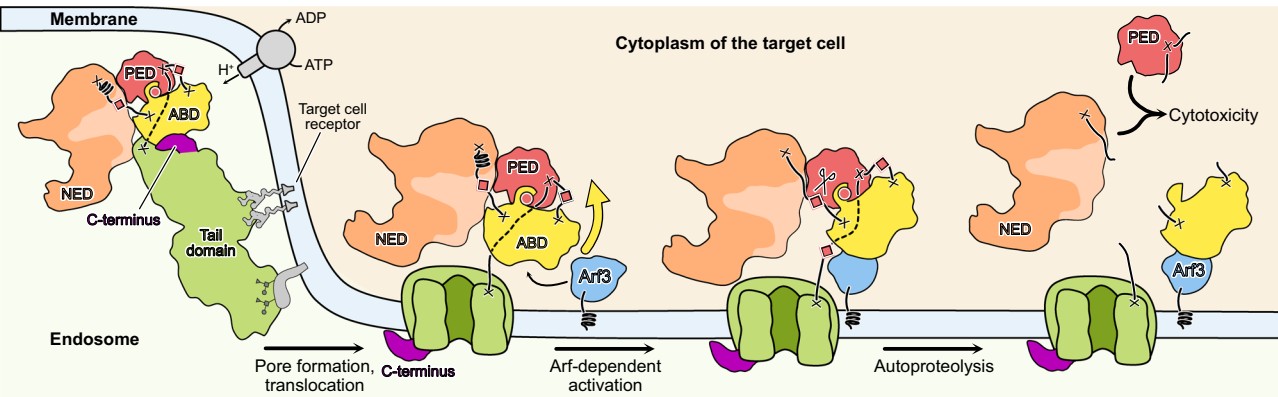

**Fig. 5 | Mechanism of action of Mcf1 toxin.** Abbreviations: see Fig. 1.

viability upon expression of recombinant genes was analyzed by serial 5-fold dilution of $OD_{600}$-normalized yeast culture[34]. To analyze protein production in yeast, the cell lysate was first prepared by treatment of cells with 0.2 M NaOH for 5 min and boiling in 4-fold Laemmli buffer[45]. Then, the extract was separated by SDS-PAGE, followed by blotting on PVDF membrane, and incubation with anti-Myc antibodies (dilution 1 to 10,000, clone 9B11 #2276, Cell Signaling Technology), anti-Mcf1 antibodies (dilution 1 to 10,000, rabbit polyclonal antibodies custom-made by Cambridge research biochemicals), ADP-ribose binding reagent (Anti-pan-ADP-ribose binding reagent, dilution 1 to 3000, MABE1016, Merck) or anti-RPS9 serum (dilution 1 to 10000, polyclonal rabbit antibodies were a generous gift of Prof. S. Rospert), and visualized using secondary anti-mouse HRP (dilution 1 to 3000, #1706516, Bio-Rad) or anti-rabbit HRP-antibodies (dilution 1 to 3000, #1706515, Bio-Rad).

## Recombinant protein production and purification

Mcf1 and its variants were expressed overnight at 22 °C in the presence of 0.02 mM IPTG in *E. coli* BL21-CodonPlus (DE3)-RIPL cells transformed with plasmids listed in Supplementary Table 2. Following the lysis by ultrasonication in buffer containing 20 mM Tris-HCl pH 8 and 100 mM NaCl, the soluble fraction was loaded onto Ni-IDA resin, washed with the lysis buffer, and eluted with the same buffer supplemented with 250 mM imidazole. The eluates were loaded on a mono Q 5/50 column and eluted with a linear gradient from 100 to 500 mM NaCl in 20 mM Tris-HCl pH 8, and stored in aliquots at −20 °C.

Arf3 with the N-terminal deletion of 17 amino acids and the mutation Q71L was expressed overnight at 21 °C in the presence of 0.02 mM IPTG in *E. coli* BL21-CodonPlus(DE3)-RIPL cells possessing the corresponding plasmid. After lysis by sonication, and centrifugation, the lysate supernatant was purified using Ni-IDA resin and a monoQ 5/50 column as described above for Mcf1. Finally, the sample was supplemented with GTP and $MgCl_2$ to concentrations of 1 and 10 mM, respectively, and stored in aliquots at −20 °C.

TccC3 ART was expressed overnight at 22 °C in the presence of 0.1 mM IPTG in *E. coli* BL21-CodonPlus (DE3)-RIPL cells containing the corresponding plasmid. Lysis by sonication in a buffer containing 20 mM Tris pH 8 and 500 mM NaCl was followed by purification on a Ni-IDA resin as described above. Lastly, the eluate was dialyzed against a buffer containing 20 mM Tris-HCl pH 8 and 150 mM NaCl, and stored in aliquots at −20 °C[31].

## Mcf1 cleavage assays

In the total volume of 6 μl, 4 μg of Mcf1 in the 20 mM Tris-HCl pH 8 and 100 mM NaCl buffer were mixed with 4 μg of Arf3 Q71L in buffer containing 20 mM Tris-HCl pH 8, 100 mM NaCl, 10 mM $MgCl_2$ and 1 mM GTP, and incubated for the indicated amount of time at 22 °C. The addition of GTP and the presence of mutation Q71L in Arf3 assured its GTP-state, which was shown to be important for interaction with the Mcf-like effector from *V. vulnificus*[14,15]. The reaction was stopped by addition of 4-fold Laemmli buffer. The samples were analyzed by SDS-PAGE and western blot with anti-FLAG (dilution 1 to 5000, clone M2, #F3165, Sigma Aldrich) or anti-His primary antibodies (dilution 1 to 3000, #H1029 Sigma Aldrich). Uncropped gels, western blots and corresponding loading controls can be found in Supplementary Fig. 11 and in the source data.

## Intoxication assays

Sf9 cells at a concentration of 1 million/ml were intoxicated with the wild-type or C1397A Mcf1 variant. If specified, Sf9 cells were treated with bafilomycin A1. Cells were imaged after 24 h incubation at 28 °C.

For the analysis of Mcf1 cleavage during intoxication, 5 ml of Sf9 cells at the concentration of 4–5 million/ml were intoxicated with 5 nM of Mcf1 and incubated at 28 °C gently shaking for the indicated amount of time. Cells were centrifuged at 1000 g, washed three times with

20 mM Tris-HCl pH 8, 100 mM NaCl buffer, and lysed by passing the suspension for 10 times through a needle with 26 gauge in diameter. The lysate was clarified by 2 min centrifugation at 1000 g, and mixed with Laemmli buffer. The samples were analyzed by SDS-PAGE, blotted onto PVDF membranes and stained with anti-Myc antibodies (dilution 1 to 2000, clone 9B11 #2276, Cell Signaling Technology).

## Mass spectrometry

To identify the cleavage site that separates the 105 and 225 kDa Mcf1 fragments, we performed mass-spectrometry analysis of the N-terminal effector polypeptide. To this end, we first co-expressed the full-length His-tagged Mcf1 and Arf3 Q71L in *E. coli* (Supplementary Fig. 4c), lysed the cells in 20 mM HEPES pH 7.5, 100 mM NaCl, and applied the supernatant to the Ni-IDA beads. After washing with lysis buffer and elution with 250 mM imidazole, we diluted the sample twice with 20 mM HEPES pH 7.5, loaded the eluate on the Mono S 5/50 column and eluted with NaCl gradient. The fractions were loaded on SDS-PAGE, and the bands corresponding to the N-terminal effector were cut out and analyzed in quadruplicates by LC-MS/MS at Proteome Factory AG. The mass-spectrometry data is available in the source data.

## ADP-ribosylation assays

30 μl of Sf9 insect cell lysate with a total protein concentration of 3 mg/ml was mixed with 3 μg of Mcf1 N-terminal effector or 2 μg of TccC3 ART. The reaction was initiated by addition of biotinylated NAD to 16.7 μM and then incubated for 30 min at 30 °C. After separation on SDS-PAGE and blotting on PVDF membrane, ADP-ribosylation was visualized by incubating the membrane with HRP-streptavidin (dilution 1 to 5000, Pierce #21130).

## Isothermal titration calorimetry (ITC)

The potential of the NED to bind nicotinamide dinucleotide ($NAD^+$) was measured by ITC with a MicroCal PEAQ-ITC (Malvern Panalytical). TccC3 from *P. luminescens* was used as a control. The measurement was performed after overnight dialysis at 25 °C in 150 mM NaCl and 20 mM Tris-HCl pH 8 buffer. 25 μM TccC3 or NED were titrated with 2.2 mM $NAD^+$. The data was analyzed using the MicroCal PEAQ-ITC Analysis software by applying a one-site-binding fit.

## Common cryo-EM techniques

All cryo-EM samples were prepared using glow-discharged copper C-flat 2–1 400 mesh grids. After application on the grid, samples were blotted at 100% humidity for 3 s (13 °C, 0 s drain time, blot force −3 to +2) with a Vitrobot Mark IV (Thermo Fisher Scientific) and plunged into liquid ethane. All datasets were collected on a Titan Krios at 300 kV using a K3 detector (Gatan) with a nominal magnification of 105,000 x in super-resolution mode (physical pixel size 0.9 Å), using the EPU software version 2.7 or 2.8 (Thermo Fisher Scientific). The defocus ranged from −1 to −2.5 μm. The preprocessing of data was carried out on-the-fly with the TranSPHIRE software (v1.4[46]), including drift correction and dose-weighing with MotionCor2 (v1.3.0[47]), as well as CTF estimation with CTFFIND4 (v4.1.13[48]).

## Cryo-EM of full-length Mcf1

3 μl of the freshly purified Mcf1 sample at a concentration of 1.1 mg/ml, supplemented with 0.002% of Tween-20, were applied onto a grid, blotted and plunged into liquid ethane.

We collected two datasets on a Titan Krios, equipped with an energy filter (GIF BioQuantum, Gatan) set to a slit width of 20 eV. Exposure time of 3 s over 60 frames resulted in an accumulated dose of 71 e⁻/Å² for the first and 69 e⁻/Å² for the second dataset. 7907 and 15,000 movies were collected for the first and the second datasets respectively. The last 5955 micrographs of the second dataset were collected with 30° stage tilt to overcome the preferred orientation of Mcf1 inside the thin vitreous ice layer.

The acquired micrographs were manually inspected, and images with large ice contaminations, excessive drift or micrographs with a resolution limit above 6 Å were discarded. Particles were picked using SPHIRE-crYOLO (version 1.6[49]), with a model trained on 10 hand-picked micrographs. The selected particles (787,549 from 7,653 micrographs of the first dataset, and 1,983,207 from 14,088 micrographs of the second dataset) were extracted with a box size of 300 pixels and used in a 3D refinement in MERIDIEN (SPHIRE package version 1.4[50]) with coarse shifts (search range 25, step range 5 pixels). The calculated projection parameters were used to recenter and re-extract properly centered particles with a box size of 320 pixels. The centered particles were 2D classified using the iterative stable alignment and clustering approach ISAC (SPHIRE version 1.4; pre-alignment was switched off and translation search range was set to 0) to remove erroneous picks.

The following processing steps were performed differently for the first and second datasets (Supplementary Fig. 1). For the first dataset, the 787,549 particles from 2D classification were used for a 3D refinement with MERIDIEN and resulted in a 4.3 Å reconstruction. Alternating particle polishing in RELION version 3.1[51] with 3D refinement in SPHIRE improved the average resolution of the reconstruction to 3.4 Å. This density, as well as the density obtained by signal subtraction of the head was used to build the C-terminal half of the protein. We noticed that the density corresponding to the N-terminal part of the toxin was of lower resolution and appeared at lower visualization thresholds. Therefore, the particles and their projection parameters were imported into RELION 3.1 and were subjected to 3D classification without image alignment ($T = 4$, two classes, 25 iterations), with a soft mask covering the N-terminal half of the protein and a reference reconstruction low-pass filtered to 20 Å. Out of 363,044 particles, only 47,522 possessed the density corresponding to the N-terminal region. The low number of such particles and their preferred orientation in the thin vitreous ice layer hindered us from obtaining a high-resolution reconstruction for robust model building of the N-terminal part of the toxin. Therefore, we acquired a second, larger dataset with 14,088 micrographs, including 5,955 collected with a stage tilt of 30°.

Applying the same pre-processing strategy as for the first dataset, 711,705 particles were selected for the first 3D refinement, which resulted in a reconstruction with an average resolution of 4.6 Å with better isotropy. To obtain a high-resolution reconstruction of the head region, we similarly performed a 3D classification in RELION with a mask covering the head region and obtained 219,318 particles. These particles were combined with the particles from the first dataset after 3D classification. The following rounds of signal subtraction, 3D classification, 3D refinement and particle polishing, resulted in the final reconstruction at a resolution of 3.6 Å, which was used to build the N-terminal part of the toxin.

The complete composite model of Mcf1 was built de novo in three steps. First, using one original and one signal subtracted reconstruction obtained from the first dataset, we built the tail domain. Secondly, we used the reconstruction from particles of the first and the second dataset to build the head domain except for residues 216–436. Finally, we took the structure of the missing residues from the Mcf1$_{C1397A \; \Delta15C}$ structure and merged them together to generate the complete Mcf1 structure. Model building was performed manually in Coot 0.8.9.2[52] or automatized by trRosetta version 3[53] and further refined in Isolde 1.4[54] and PHENIX version 1.16[55].

The composite cryo-EM density for Fig. 1a was postprocessed using DeepEMhancer[56].

## Cryo-EM of Mcf1$_{C1397A \; \Delta15C}$

Freshly purified protein was diluted to 0.7 mg/ml (2.2 μM) and mixed with Tween-20 to a final concentration of 0.008% to improve ice quality. 3 μl of the sample were applied onto a grid, blotted and plunge-frozen in liquid ethane.

The dataset was collected using a Titan Krios, equipped with a postcolumn energy filter (slit width of 15 eV). Image stacks with 60 frames were collected with a total exposure time of 3.5 s and a total dose of 60.85 e$^-$/Å$^2$. 9032 images were acquired and 6987 of them were used for processing. After on-the-fly preprocessing, particle picking was performed using SPHIRE-crYOLO version 1.7. 2,202,784 particles were then extracted in RELION 3.1.0 with a pixel size of 1.8 Å and a box size of 150 pixels, and transferred into CryoSPARC version 4.1.0[57]. After 3 independent 2D classifications (in 150, 175 or 200 classes) and ab initio 3D refinement (in 3 classes), we ended up with 544,954 particles and an ab initio 3D model, which we later refined to a 3.72 Å reconstruction by non-uniform refinement. Then, we reextracted and unbinned particles to a pixel size of 0.9 Å, performed two series of particle polishing in RELION 3.1.0, and finally computed a 3.08 Å resolution reconstruction using local refinement with a mask covering the N-terminal effector and the PED.

In the next step, we took coordinates of the particles used in the reconstruction and used them to train a crYOLO picking model. Picking resulted in 2,181,255 particles, which were extracted in RELION 3.1.0 with a pixel size of 1.8 Å and a box size of 150 pixels and transferred into CryoSPARC version 4.1.0. After 2D classification and removal of junk particles, we performed a non-uniform 3D refinement with 904,685 particles, followed by a round of alignment-free 3D classification in 10 classes. Particles from class 7, which showed a complete ABD, were unbinned, and subjected to two rounds of non-uniform refinement in CryoSPARC and polishing in RELION. Finally, after the last clean-up by alignment-free 3D classification, we ended up with 53,952 particles, refining them to a 3.46 Å resolution reconstruction covering the NED, PED and ABD of Mcf1.

To build the atomic models, we rigid-body fitted the previously built structure of the Mcf1 head into the density in UCSF Chimera version 1.14[58], built the missing amino acids and refined the structure in Isolde version 1.4[54], trRosetta software version 3[53] and Phenix[55] version 1.17.

## Cryo-EM of the Mcf1$_{C1397A \; \Delta15C}$-Arf3 complex

Freshly purified Mcf1$_{C1397A \; \Delta15C}$ at 2 mg/ml (6.2 μM) was mixed with a 5-fold molar access of Arf3 Q71L, diluted to 0.5 mg/ml, and mixed with Tween-20 to the final concentration of 0.006% to improve ice quality. The sample was applied onto a grid, blotted and plunge-frozen in liquid ethane.

The dataset was collected using a Titan Krios, equipped with a C$_s$-corrector and a postcolumn energy filter (slit width of 15 eV). Image stacks with 60 frames were collected with the total exposure time of 2 s and a total dose of 62.6 e$^-$/Å$^2$. 18,671 images were acquired and 15,693 of them were used for processing. After on-the-fly pre-processing, particle picking was performed using crYOLO version 1.7. 1,491,851 particles were then extracted in RELION 3.1.0 with a pixel size of 1.8 Å and a box size of 150 pixels, and transferred to CryoSPARC version 4.1.0. After 3 independent 2D classifications (in 100, 150 or 200 classes) and ab initio 3D refinement (in 3 classes), we ended up with 770,049 particles and an ab initio 3D model, which we later refined to a 4.3 Å reconstruction by non-uniform refinement, followed by a round of alignment-free 3D classification in 10 classes. Then, we took class 5 with the largest number of particles and proceeded with a non-uniform refinement and unbinning to a pixel size of 0.9 Å. The particles were refined once again in the original pixel size in CryoSPARC and polished in RELION. The final 3D refinement in CryoSPARC resulted in a reconstruction of 4 Å resolution, which we used for model building.

To build the atomic model of the Mcf1$_{C1397A \; \Delta15C}$-Arf3 complex, we rigid-body fitted the Mcf1$_{C1397A \; \Delta15C}$ and Arf3 models from PDB 6II6[15] into the density, and then performed flexible fitting using iMODFIT Chimera plugin[59], version 1.2. The resulting model was refined in Isolde version 1.4[54] and Phenix version 1.17[55].

**Reporting summary**

Further information on research design is available in the Nature Portfolio Reporting Summary linked to this article.

## Data availability

The coordinates for the cryo-EM structures of the full-length Mcf1, Mcf1C1397A Δ15C, and the Mcf1C1397A Δ15C-Arf3 complex have been deposited in the Electron Microscopy Data Bank under accession numbers EMD-17440 (composite map; original and low-resolution consensus maps – EMD-17437, 17438, 17437, 17450), EMD-17436 and EMD-17435. The corresponding molecular models have been deposited at the wwPDB with accession codes PDB 8P52, 8P51 and 8P50. The raw data generated during the current study are available from the corresponding author on request. Uncropped gels, western blots and corresponding loading controls can be found in the Source data and Supplementary Fig. 11. We used the following previously published structures: 6II6, 7POG, 7V1N, 7RT7, 3TU8, 6SUS, 3O4J, 6QK7. Source data are provided with this paper.

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

## Acknowledgements

We thank S. Bergbrede and M. Hülseweh for the excellent technical assistance, S. Rospert for providing us with anti-RPS9 serum, A. Rai for advice on biochemistry of Arf proteins and providing us with Arf6 for initial tests, R. Gasper-Schönenbrücher for assistance during the ITC measurement, W. Oosterheert, C. Gatsogiannis, O. Sitsel and Y. Belyi for fruitful discussions. This work has been funded by the Max Planck Society (S.R.). A.B was supported by an EMBO long-term fellowship and a stipend of the Humboldt foundation.

## Author contributions

S.R. and A.B. designed the project. A.B and P.Heilen performed wet lab experiments. P.Hagel performed initial cryo-EM experiments. A.B. prepared cryo-EM specimens. O.H. collected cryo-EM data. A.B. and P.Heilen analyzed cryo-EM data and built atomic models. A.B., P.Heilen and S.R. wrote the manuscript with input from all co-authors.

## Funding

## Competing interests

The authors declare no competing interests.
