## [Peer Review File · Nature Communications]

Reviewers' Comments:

Reviewer #1:

Remarks to the Author:

This manuscript describes the structure of a bacterial toxin called Makes Caterpillars floppy 1 (Mcf1). As the first structural report for this type of toxin, there is a lot of novelty, both in terms of the unique structural elements and the elements with structural homology to other well characterized toxins, in particular, the large clostridial toxins TcdA and TcdB. The authors have performed a rigorous structural analysis, using cryo-EM to determine structures of Mcf1 alone, and in complex with Arf1. They pair the structural work with autoprocesing assays and a clever yeast toxicity assay. The experimental work is rigorous and clear.

There are places however where I think the authors overstate what has been shown, where the analysis could be strengthened, or where the writing could be clearer.

1. At the end of page 2, top of page 3, the authors state that Mcf1 has a hydrophobic region similar to that observed in TcdB. It is confusing that the authors don't also list TcdA, since later, the sequence comparison they show is with TcdA. I thought the authors might be making a distinction between TcdA and TcdB when, in fact, both appear similar.
2. On page 3, the authors state that the tail consists of two translocation domains and three receptor binding domains but do not provide evidence that these structures perform these functions.
3. Supplemental Figure 2 would be more compelling if the authors can provide an rmsd for the structures being compared. In some places the structural similarities are clear, but it would be helpful to color the region of the NED that aligns to RHSP2.
4. The authors state: 'Since the other domains are also structurally reminiscent of the receptor-binding domains of Tc toxins⁴ and diphtheria toxins²⁴, we called these domains receptor-binding domains 1-3 (RBD1-3) (Fig. 1c and d).' This is an important point if they want to infer that these domains have potential roles in receptor binding. These examples should be shown in Supplemental Figure 2 along with rmsds for the regions being compared.
5. The description of the $\Delta 15C$ mutant on page 9 would be easier to follow if there were a figure that defined these residues by color or sequence number. That said, it is a leap to say that the regulation comes from conformational changes associate with membrane insertion and conformational changes when this is not specifically tested. Qualifying that this as a hypothesis would be preferred.
6. Similarly, the authors do not explicitly show that one of the autoprocesing sites is inaccessible due to membrane insertion. This statement should be presented as a hypothesis.
7. In addition to the overall structural comparison, it could prove helpful to analyze the putative ADP-ribosyltransferase fold for the predicted or actual capacity to bind NAD.
8. 'On a molecular level, we suggest that Arf3 binding strains the helices of the activator-binding domain and drives the movement of its N-terminal part (Fig. 4b and c). This movement is possible due to a hook-loop interaction between the activator-binding domain and the protease effector domain, organized by arginine 957 of the activator-binding domain and the loop 1291-1298 of the protease effector domain (Supplementary Fig. 6b). This structural feature allows the activator-binding domain to swing away from the N-terminal effector, and to pull the cleavable linker toward the catalytic center of the protease effector domain for proteolysis (Fig. 4c).'
- I was not able to follow this. Can the figures be improved? Labeling the hook/loop would help. And clearly showing the two positions for the cleavable linker.
9. Can the figure legend for Figure 4 be improved to define the differences between the three Mcf1-Arf3 complex structures? Are these from different samples? Classes? Where is the hook-loop?
10. 'We propose that two positively charged regions on the side and the bottom of the toxin (Supplementary Fig. 9) form initial contacts with the negatively charged surface of the membrane.'
- Unclear what this is based on.
11. Methods state that 1 mM GTP was used in Mcf1 cleavage assays. Why? Was this required? If so, this should be clearly stated.

Reviewer #2:

Remarks to the Author:

This is an elegant structural and functional study of the “makes caterpillars floppy” toxin from *Photobacterium luminescens*, one of many insecticidal toxins produced by this organism, and which shares similarities with other, significant toxin systems including the MARTX toxins and the glucosylating toxins of *Clostridia*. The manuscript presents the structure of this toxin, and in a stepwise manner deduces some interesting elements of its function which overall presents an insightful picture of the functional mechanism that underpins its virulence.

For the most part, the data are analysed appropriately and the methodology is both sound and standard for the field. The conclusions are also, for the most part, supported by the evidence provided in the manuscript, or reasonable interpretation of these evidences. There are places in the manuscript where conclusions are drawn prematurely. Similarly, I have some concerns that there are places in the manuscript where conclusions are drawn in an absolute context, when the evidence perhaps justifies a more conservative conclusion. Regardless, I feel that the manuscript offers more than sufficient novelty and significance to warrant publication, with due consideration having been given to the comments I have included below.

* Last paragraph of page 2: This section summarises the identifiable homology between Mcf1 and other characterised systems. It would be useful in my opinion if a schematic was included somewhere in the manuscript that summarised this visually. Including a comparison in this schematic to other prominent toxins referred to in the manuscript, including TcdA, TcdB and Vibrio MARTX would greatly assist the reader in understanding the context and novel aspects of this structure.

* First paragraph of page 3: It seems unusual here to mention that Mcf1 does not contain a CROPs domain. The way the last sentence of this paragraph is written, it makes it sound as if this is the only difference between Mcf1 and TcdA/B. There are other differences apparent in the structures as well, so why only mention this one specifically? On page 14, paragraph 2 of the discussion, a similar statement is made, but instead highlights all the structural differences (not just the CROPs domain). It would therefore be reasonable to say something similar in the first instance, or to remove the reference to the CROPs domain and just say Mcf1 and TcdA/B are similar.

* The similarity between what the authors call the PED and the Mcf-like domain of MARTX is something that had been inferred from several previously published studies. The authors cite a 2023 preprint from Herrera et al. when acknowledging this similarity, but other papers (including reference 17 cited by the authors elsewhere) also noted this and could be included here. In general, I think the authors could do a better job of acknowledging previous work that has inferred the presence of certain subdomains and associated functionalities, albeit largely on the basis of sequence and without any accompanying structural context.

* It is also not clear to me why the authors felt it necessary to rename this particular domain in calling it a protease effector domain (PED) when other toxins with similar domains refer to it as a MCF-like domain, on the basis of its similarity to this exact domain? It would make more sense to simply call it the MCF domain?

* First paragraph of page 6: It is not clear to me on what basis the authors arrived at the conclusion that RBD1-3 are similar to the RBDs of Tc and diphtheria toxins? The data in Supp Fig. 2 identify alternative structural homologues and there is no evidence presented here or elsewhere that I could find that these were identified in e.g. a fold search or by some other method? The authors should clarify the basis for their conclusion that these specific molecules were identified as being “structurally reminiscent”. RBDs from Tc toxins and diphtheria are merely immunoglobulin-like beta sandwich domains that are much more widely distributed than in just these two toxin systems so drawing attention to these two alone seems unjustified on the basis of the data presented and may inadvertently suggest a closer structural relationship that what the authors intended to infer.

* Second paragraph of page 6: the phrase “structurally correctly positioned” is an unusual one to use here. Why not just correctly positioned, or structurally conserved?

* With regard to this same sentence, the authors conclude, on the basis that the catalytic triad is structurally conserved, that Mcf1 functions as a protease. This conclusion is in my opinion premature at this point in the manuscript. Up to this point, no evidence is presented that Mcf1 is a functional protease, and indeed this observation seems to be the basis of the hypothesis that leads to them evaluating its *potential* protease activity, which they do subsequently go on to establish. It would be better to state here that the catalytic triad is *suggestive* of it being a protease.

* Again in the subsequent paragraph on this page, the statement that Mcf1 is cleaved in at least two places is premature at this stage, since no evidence is presented at this point about the physiological or functional relevance of the second (and any subsequent cleavage events). It would be appropriate to qualify this conclusion at this point in the manuscript.

* In several places, the authors highlight differences between Mcf1 and other toxins (e.g. page 6, paragraph 3; page 9, paragraph 3). I believe these statements are far too general in nature and are instead intended to distinguish Mcf1 from the structurally related toxins discussed in this manuscript. The authors should make this clear in their wording. Similarly (page 11, paragraph 1) refers to a feature the authors claim may be a general feature of Photorhabdus toxins, without being clear whether they mean all Mcf-like toxins from photorhabdus, or all photorhabdus toxins in general.

* Page 7, paragraph 1: It is stated that Mcf1 does not have an IP-6 binding site. Did the authors test for this using an appropriate binding assay? Or is this just something that is inferred based on the structural data? If the latter, it would be more appropriate to say there is no *apparent* or *obvious* IP-6 binding site.

* Regarding the C-terminal regulatory element, again I believe the authors present a premature conclusion in this regard when they state on page 9, paragraph 2 "This result demonstrates that the C-terminus of Mcf1 is a regulatory element that prevents autoprocessing...". The single piece of evidence presented up to this point is, in my opinion, merely suggestive of such a function. Later in the manuscript, the authors present much more convincing evidence. The explanation for why the C-terminal deletion exhibits enhanced proteolysis in the last paragraph on page 12 is much more justifiable. The authors should therefore temper or qualify their conclusion earlier in the manuscript with regard to this region of the proteins putative function. Another point regarding this structural feature is that it would be nice if the C-terminus was clearly highlighted in Figure 1 or at another appropriate point in the manuscript.

* Page 9, paragraph 4: Since the authors could not identify clearly the location of cleavage sites 2 and 3, is it possible that this is the result of non-specific proteolysis?

* Page 9, paragraph 5: The statement that TD1 is membrane-bound during intoxication is neither supported by evidence or references and it should therefore be made clear that this is an assumption. The immediately following sentence which starts "This result demonstrates" also seems to suggest that this is something for which the authors provide evidence. There is no direct evidence for this that I can see anywhere in the manuscript, and so again the sentence should be reworded to indicate that "such a scenario would corroborate the idea..." or similar.

* I found it intriguing that the authors mentioned similarity between a region of the ABD and the BH3 domain of apoptotic proteins. This was not highlighted in the structure, and indeed was not mentioned again until page 10, paragraph 4, where the authors ultimately provide evidence that the proteolytic fragment containing this sequence has no apparent apoptotic activity. Again, in the interest of being concise, it might be better to remove mention of this from the introduction and simply state on page 10 that, despite sharing similarity with BH3-like proteins and Burkholderia lethal factor, there was no detectable cytotoxicity associated with this fragment. On this point, the authors reached the conclusion that the domain is therefore inactive, but retained for structural purposes, and while I believe it is completely justified to speculate on this point, it should be made clear that this is speculation, especially since it is based upon only a single negative result, and that the activity was tested in yeast cells, which are not the native target of Mcf.

* The described blurry-ness of the 2D class averages in Supp Fig.7 and Supp Fig. 8 is very hard to see at the scale that these are reproduced. A similarly comment could be made about the 2D classes they are supposed to be compared to in Supp Fig. 1. Given their importance, I would suggest showing a few, representative, enlarged 2D classes that highlight this comparison.

* Page 13, paragraph 1: The statement that N-terminal effector domain and protease effector domain are structurally rigid. Is there evidence for this, or is this an assumption to facilitate the subsequent mechanistic modelling? It would also be appropriate to refer to these domains by their abbreviated names, given these were defined earlier in the manuscript.

* Page 13, paragraph 2: Again, is this speculation or based on evidence? If the latter, please highlight the evidence more clearly. If it is speculation, please make it clear that this is the case.

* Following on, Page 13, paragraph 3: "To test this mechanism" – I believe a more appropriate terminology would be hypothesis rather than mechanism. The mutants described in the following section indeed test the hypothesis and help to establish the mechanism.

* Page 14, discussion paragraph 2: The receptor binding domains should be described as putative, since no receptor binding function has been directly demonstrated.

* Page 15, paragraph 1: The last sentence of this paragraph makes a rather sizeable logic jump to suggest that there are major applications for Mcf1 in cancer and gene therapy, without any other related discussion or explanation of how this might be the case. The authors should provide more context around this, or ideally remove the sentence altogether as it seems to be to be an unnecessary exaggeration of significance.

* Page 15, paragraph 2: The authors make the statement that their structural and functional data "together with previous analyses by other groups" all them to propose a mechanism, but again I believe the discussion of other relevant literature is a little underdone here. The sentence itself is made without citing any relevant literature and at the very least the authors should cite relevant literature that supports their proposed mechanism of action here. Indeed this entire paragraph outlining the proposed mechanism of action has no citations to any other published work, which makes it hard to appreciate the similarities to other systems, upon which the authors have presumably based their mechanism of action (at least in part). Ideally, a fuller discussion of the relevant literature would also accompany this section of the paper.

* Finally, and along these same lines, I was surprised not to find more discussion or comparison to other potentially related systems regarding certain aspects of the structure. To name a few:

- There is only a very limited mention of the PaTox toxin, which the authors flag as being similar to the protease effector domain.
- The Adenylate cyclase toxins – which according to data presented in Supp Fig. 2 is a close structural homologue of (at least) two of the domains in the C-terminal region – is not mentioned anywhere in the manuscript that I could find?
- The similarity to the MARTX system of Vibrio I believe is also slightly under-discussed, given that the authors basically posit that Mcf is a scramble between the Clostridial TcdA/B systems and the Vibrio MARTX system. It might be reasonable to include a comparison to MARTX alongside Supp Fig. 3, for example, given this statement?

Minor points

* The phrase "for the first time" in the last sentence of the introduction is an unnecessary inclusion. The merits of the manuscript still stand even if this is statement is omitted. Since many journals prefer that authors don't include unqualified claims of priority, it might be better to remove this phrase.

* Paragraph 3, page 4. There should be a comma after "Therefore" at the start of the second sentence.

* Second paragraph, page 6: The last sentence should be rephrased – the words "results in Mcf1 to remain uncleaved" should simply say "remains uncleaved".

* Paragraph 2, page 7: the wording "resulted indeed" in the last sentence would be better phrased "indeed resulted"

* Figure 2 is a very dense figure and it took me a long time looking at this to figure out everything that was presented here. I would encourage the authors to consider if there is a way that this information can be presented more clearly or intuitively, although I confess I don't have any good suggestions for how to do this myself!

* Page 9, paragraph 4: Specify is an unusual term to use here – maybe locate or identify would be better?

* Page 10, paragraph 3: The last two sentences of this paragraph could be summarised much more succinctly by saying that despite sharing structural homology with ADP-ribosyltransferases, the Mcf1 NTE domain showed no detectable ADP-ribosyltransferase activity in yeast or insect cell assays. The rest of these sentences are simply saying that the control experiment worked.

* Page 14, paragraph 1 of the Discussion: The opening paragraph should be rephrased to make it clear that the authors are not suggesting Mcf1 is a major cause of health care-associated illnesses.

Point-to-point response to the reviewers' comments

We thank the reviewers for their positive and constructive feedback, which aided us to further improve the manuscript. Below we include our detailed response to each point raised. The changes are highlighted in yellow in the revised text.

Reviewer #1 (Remarks to the Author):

This manuscript describes the structure of a bacterial toxin called Makes Caterpillars floppy 1 (Mcf1). As the first structural report for this type of toxin, there is a lot of novelty, both in terms of the unique structural elements and the elements with structural homology to other well characterized toxins, in particular, the large clostridial toxins TcdA and TcdB. The authors have performed a rigorous structural analysis, using cryo-EM to determine structures of Mcf1 alone, and in complex with Arf1. They pair the structural work with autoprocessing assays and a clever yeast toxicity assay. The experimental work is rigorous and clear.

We thank the reviewer for the very positive feedback on our manuscript. We are especially pleased that the reviewer appreciates the novelty of our study.

There are places however where I think the authors overstate what has been shown, where the analysis could be strengthened, or where the writing could be clearer.

1.1 At the end of page 2, top of page 3, the authors state that Mcf1 has a hydrophobic region similar to that observed in TcdB. It is confusing that the authors don't also list TcdA, since later, the sequence comparison they show is with TcdA. I thought the authors might be making a distinction between TcdA and TcdB when, in fact, both appear similar.

We thank the reviewer for noticing this. We made changes in the text and now write about both TcdA and TcdB in the introduction:

“Third, Mcf1 possesses a highly hydrophobic sequence similar to that in the translocation region of the *Clostridioides difficile* toxins A and B (TcdA, TcdB), suggesting that Mcf1 can undergo a conformational change and penetrate the endosomal membrane, similar to these toxins.”

1.2. On page 3, the authors state that the tail consists of two translocation domains and three receptor binding domains but do not provide evidence that these structures perform these functions.

Our naming of translocation and receptor-binding domains was indeed based exclusively on the structural and sequential similarity of Mcf1 to other toxins. We agree with the reviewer that such data does not provide direct evidence of the functions of these domains. Therefore, agreeing with this and the following remarks of both reviewers, we decided to call these domains **putative** translocation domains and **putative** receptor-binding domains.

1.3. Supplemental Figure 2 would be more compelling if the authors can provide an rmsd for the structures being compared. In some places the structural similarities are clear, but it would be helpful to color the region of the NED that aligns to RHSP2.

We agree with the reviewer. We now provide rmsds for the structures being compared and have colored the region of the NED that aligns to RHSP2 in Supplementary Figure 2.

1.4. The authors state: ‘Since the other domains are also structurally reminiscent of the receptor-binding domains of Tc toxins⁴ and diphtheria toxins²⁴, we called these domains receptor-binding domains 1-3 (RBD1-3) (Fig. 1c and d).’ This is an important point if they want to infer that these domains have potential roles in receptor binding. These examples should be shown in Supplemental Figure 2 along with rmsds for the regions being compared.

We agree with the reviewer that distant structural similarity does not necessarily imply the identical function. Therefore, we specified the comparison, referencing the structures from Supplementary Fig. 2 that are now complemented by rmsds.

“The tail of Mcf1 ends with three β -strand-rich domains. The latter one as well as translocation domain 2 show structural similarity to the β -roll structures of RTX toxins without sharing their consensus sequence and characteristic Ca^{2+} binding⁵⁷. Since the last domain also shares structural similarity to a region of *C. difficile* toxins that has been connected to receptor binding (Supplementary Fig. 2), we called these domains putative receptor-binding domains 1-3 (RBD1-3) (Fig. 1c and d).”

1.5. The description of the \square 15C mutant on page 9 would be easier to follow if there were a figure that defined these residues by color or sequence number. That said, it is a leap to say that the regulation comes from conformational changes associate with membrane insertion and conformational changes when this is not specifically tested. Qualifying that this as a hypothesis would be preferred.

Agreeing with the reviewer, we color-coded the C-terminal region of the toxin in Fig. 1 and changed the text as follows:

“We hypothesized that membrane insertion and translocation of Mcf1 during intoxication of insect cells would inevitably result in conformational changes at the neck region, where the C-terminus of the protein makes strong hydrophobic interactions with the head of Mcf1 (Fig. 1g).”

“This result demonstrates that the C-terminus of Mcf1 is a potential regulatory element that prevents autoprocessing of Mcf1 prior to its translocation into the target cell.”

1.6. Similarly, the authors do not explicitly show that one of the autoprocessing sites is inaccessible due to membrane insertion. This statement should be presented as a hypothesis.

We changed our text as follows:

“In contrast to the *in vitro* conditions, where the protease effector domain can access this region and excise the putative translocation domain 1, during cell intoxication this domain is likely membrane-bound and therefore inaccessible to the protease. Such a scenario would further corroborate the idea that the conformational change during membrane penetration precedes toxin activation inside the host cells.”

1.7. In addition to the overall structural comparison, it could prove helpful to analyze the putative ADP-ribosyltransferase fold for the predicted or actual capacity to bind NAD.

We performed isothermal titration calorimetry (ITC) measurements comparing the binding of NAD⁺ to TccC3, which is a known ADP-ribosyltransferase and to the N-terminal effector domain of Mcf1. While we could measure that NAD⁺ binds to TccC3 with an affinity of ~ 100 μM, NAD⁺ did not bind to the N-terminal effector domain of Mcf1. We included the data in Supplementary Fig. 6, changed the main text accordingly and added a respective section in the methods section.

1.8. ‘On a molecular level, we suggest that Arf3 binding strains the helices of the activator-binding domain and drives the movement of its N-terminal part (Fig. 4b and c). This movement is possible due to a hook-loop interaction between the activator-binding domain and the protease effector domain, organized by arginine 957 of the activator-binding domain and the loop 1291-1298 of the protease effector domain (Supplementary Fig. 6b). This structural feature allows the activator-binding domain to swing away from the N-terminal effector, and to pull the cleavable linker toward the catalytic center of the protease effector domain for proteolysis (Fig. 4c).’

I was not able to follow this. Can the figures be improved? Labeling the hook/loop would help.

And clearly showing the two positions for the cleavable linker.

In order to improve the illustration, we moved parts of Supplementary Fig 6 to Fig 4 d and e and added a schematic illustration focused on the hook/loop.

1.9. Can the figure legend for Figure 4 be improved to define the differences between the three Mcf1-Arf3 complex structures? Are these from different samples? Classes? Where is the hook-loop?

The Mcf1-Arf3 complex structures in Figure 4 are 3D classes from the same sample. As proposed by the reviewer, we have added additional information to the legend of Figure 4 and added a schematic highlighting the hook-loop.

1.10. ‘We propose that two positively charged regions on the side and the bottom of the toxin (Supplementary Fig. 9) form initial contacts with the negatively charged surface of the membrane.’ Unclear what this is based on.

We agree with the reviewer and changed our text as follows:

The surface of the tail of Mcf1 is mostly negatively charged (Supplementary Fig. 10), suggesting that the toxin does not interact directly with the membrane but is directed there by interaction with a receptor located in the membrane.

1.11. Methods state that 1 mM GTP was used in Mcf1 cleavage assays. Why? Was this required? If so, this should be clearly stated.

Following the described cleavage protocols for the Mcf-like effector from *Vibrio vulnificus* (Lee et al., PNAS 2019; Herrera et al., Cell Microbiol 2020), we used GTP-bound Arf3 and therefore added GTP in our Mcf1 cleavage assays. We added a sentence in the methods section to better explain it.

“Addition of GTP and the presence of mutation Q71L in Arf3 assured its GTP-state, which was shown to be important for interaction with the Mcf-like effector from *V. vulnificus* (Lee et al., PNAS 2019; Herrera et al., Cell Microbiol 2020)”.

Reviewer #2 (Remarks to the Author):

This is an elegant structural and functional study of the “makes caterpillars floppy” toxin from *Photobacterium luminescens*, one of many insecticidal toxins produced by this organism, and which shares similarities with other, significant toxin systems including the MARTX toxins and the glucosylating toxins of Clostridia. The manuscript presents the structure of this toxin, and in a stepwise manner deduces some interesting elements of its function which overall presents an insightful picture of the functional mechanism that underpins its virulence.

For the most part, the data are analysed appropriately and the methodology is both sound and standard for the field. The conclusions are also, for the most part, supported by the evidence provided in the manuscript, or reasonable interpretation of these evidences. There are places in the manuscript where conclusions are drawn prematurely. Similarly, I have some concerns that there are places in the manuscript where conclusions are drawn in an absolute context, when the evidence perhaps justifies a more conservative conclusion. Regardless, I feel that the manuscript offers more than sufficient novelty and significance to warrant publication, with due consideration having been given to the comments I have included below.

We thank the reviewer for the very positive opinion on our manuscript.

2.1 Last paragraph of page 2: This section summarises the identifiable homology between Mcf1 and other characterised systems. It would be useful in my opinion if a schematic was included somewhere in the manuscript that summarised this visually. Including a comparison in this schematic to other prominent toxins referred to in the manuscript, including TcdA, TcdB and Vibrio MARTX would greatly assist the reader in understanding the context and novel aspects of this structure.

Following the suggestion of the reviewer, we have now added a schematic (Supplementary Fig. 3e) illustrating homologies between Mcf1 and other toxins.

2.2 First paragraph of page 3: It seems unusual here to mention that Mcf1 does not contain a CROPs domain. The way the last sentence of this paragraph is written, it makes it sound as if this is the only difference between Mcf1 and TcdA/B. There are other differences apparent in the structures as well, so why only mention this one specifically? On page 14, paragraph 2 of

the discussion, a similar statement is made, but instead highlights all the structural differences (not just the CROPs domain). It would therefore be reasonable to say something similar in the first instance, or to remove the reference to the CROPs domain and just say Mcf1 and TcdA/B are similar.

The absence of a CROPs domain was the most prominent for us when comparing the structures for the very first time. However, agreeing with the reviewer, to make the description clearer, we shortened the last sentence of the paragraph to the following.

“This analysis revealed that Mcf1 is built in a similar fashion to TcdA and TcdB toxins from *C. difficile*.”

2.3 The similarity between what the authors call the PED and the Mcf-like domain of MARTX is something that had been inferred from several previously published studies. The authors cite a 2023 preprint from Herrera et al. when acknowledging this similarity, but other papers (including reference 17 cited by the authors elsewhere) also noted this and could be included here. In general, I think the authors could do a better job of acknowledging previous work that has inferred the presence of certain subdomains and associated functionalities, albeit largely on the basis of sequence and without any accompanying structural context.

We thank the reviewer for this comment. We have not noticed that the similarity between Mcf-like domain of MARTX and PED of Mcf1 had been discussed before. We have now added a citation to Waterfield et al., FEMS Microbiology Letters 2003.

2.4 It is also not clear to me why the authors felt it necessary to rename this particular domain in calling it a protease effector domain (PED) when other toxins with similar domains refer to it as a MCF-like domain, on the basis of its similarity to this exact domain? It would make more sense to simply call it the MCF domain?

We called this domain a protease effector domain (PED) to stress that this domain is not only a protease that performs autoproteolytic cleavage of the toxin, but also a toxic effector (Fig. 3 a and d). Therefore, we kindly disagree with the reviewer and would like to keep the protease effector domain name. Unlike ‘Mcf domain’, our name gives the reader an idea of the function of the domain.

2.5 First paragraph of page 6: It is not clear to me on what basis the authors arrived at the conclusion that RBD1-3 are similar to the RBDs of Tc and diphtheria toxins? The data in Supp Fig. 2 identify alternative structural homologues and there is no evidence presented here or elsewhere that I could find that these were identified in e.g. a fold search or by some other method? The authors should clarify the basis for their conclusion that these specific molecules were identified as being “structurally reminiscent”. RBDs from Tc toxins and diphtheria are merely immunoglobulin-like beta sandwich domains that are much more widely distributed than in just these two toxins systems so drawing attention to these two alone seems unjustified on the basis of the data presented and may inadvertently suggest a closer structural relationship that what the authors intended to infer.

We thank the reviewer for bringing this up. We agree that our phrasing ‘structurally reminiscent’ suggests a higher similarity between receptor-binding domains of Mcf1 and other toxins that it actually is. Therefore, we changed the text as follows:

“The tail of Mcf1 ends with three β -strand-rich domains. The latter one as well as translocation domain 2 show structural similarity to the β -roll structures of RTX toxins without sharing their consensus sequence and characteristic Ca^{2+} binding⁵⁷. Since the last domain also shares structural similarity to a region of *C. difficile* toxins that has been connected to receptor binding (Supplementary Fig. 2), we called these domains putative receptor-binding domains 1-3 (RBD1-3) (Fig. 1c and d).”

2.6 Second paragraph of page 6: the phrase “structurally correctly positioned” is an unusual one to use here. Why not just correctly positioned, or structurally conserved?

We have changed the “structurally correctly positioned” to “correctly positioned”.

2.7 With regard to this same sentence, the authors conclude, on the basis that the catalytic triad is structurally conserved, that Mcf1 functions as a protease. This conclusion is in my opinion premature at this point in the manuscript. Up to this point, no evidence is presented that Mcf1 is a functional protease, and indeed this observation seems to be the basis of the hypothesis that leads to them evaluating its *potential* protease activity, which they do subsequently go on to establish. It would be better to state here that the catalytic triad is *suggestive* of it being a protease.

We agree with the reviewer and have changed the sentence to:

“The structure revealed that Mcf1 possesses a complete protease domain. It has a typical α/β hydrolase fold with a correctly positioned catalytic triad comprising a cysteine, histidine and aspartic acid (Fig. 3a) indicating that it likely functions as a protease.”

2.8 Again in the subsequent paragraph on this page, the statement that Mcf1 is cleaved in at least two places is premature at this stage, since no evidence is presented at this point about the physiological or functional relevance of the second (and any subsequent cleavage events). It would be appropriate to qualify this conclusion at this point in the manuscript.

Indeed, in the following paragraph we discussed the cleavage of Mcf1 in cells (Fig. 2a). We found that the protein is proteolyzed into the 105 N-terminal and 150 C-terminal fragments. Considering that summing 105 and 150 kDa does not result in 330 kDa of the full-length toxin, we concluded that Mcf1 is cleaved at least in two places. Therefore, we kindly disagree with the reviewer and believe that our conclusion appropriately corresponds to the presented experiment.

2.9 In several places, the authors highlight differences between Mcf1 and other toxins (e.g. page 6, paragraph 3; page 9, paragraph 3). I believe these statements are far too general in

nature and are instead intended to distinguish Mcf1 from the structurally related toxins discussed in this manuscript. The authors should make this clear in their wording. Similarly (page 11, paragraph 1) refers to a feature the authors claim may be a general feature of *Photorhabdus* toxins, without being clear whether they mean all Mcf-like toxins from *photorhabdus*, or all *photorhabdus* toxins in general.

We thank the reviewer for raising these points. We changed the wording of the following sentences to make it clearer.

“This is clearly different from *C. difficile* TcdA and TcdB that have a single cleavage site.” (Page 6, paragraph 3).

“In our cell intoxication and *in vitro* cleavage experiments, we detected either two or multiple fragments of Mcf1, which is surprising given the fact that *C. difficile* TcdA and TcdB are only cleaved at one position.” (Page 9, paragraph 3).

“Therefore, a dual role of protease domains might be a key feature of Mcf-like toxins from *Photorhabdus* that allows to increase their overall potency.” (Page 11, paragraph 1).

2.10 Page 7, paragraph 1: It is stated that Mcf1 does not have an IP-6 binding site. Did the authors test for this using an appropriate binding assay? Or is this just something that is inferred based on the structural data? If the latter, it would be more appropriate to say there is no *apparent* or *obvious* IP-6 binding site.

We made the statement of absence of IP-6 binding site based on the structural data. Agreeing with the reviewer, we changed the following sentence.

“However, based on our structural data, Mcf1 does not have an apparent IP6-binding site and its catalytic center does not require reorganization since all three residues of the catalytic triad are in 3 Å proximity.”

2.11 Regarding the C-terminal regulatory element, again I believe the authors present a premature conclusion in this regard when they state on page 9, paragraph 2 “This result demonstrates that the C-terminus of Mcf1 is a regulatory element that prevents autoprocessing...”. The single piece of evidence presented up to this point is, in my opinion, merely suggestive of such a function. Later in the manuscript, the authors present much more convincing evidence. The explanation for why the C-terminal deletion exhibits enhanced proteolysis in the last paragraph on page 12 is much more justifiable. The authors should therefore temper or qualify their conclusion earlier in the manuscript with regard to this region of the proteins putative function. Another point regarding this structural feature is that it would be nice if the C-terminus was clearly highlighted in Figure 1 or at another appropriate point in the manuscript.

We agree with the reviewer on this point. We tuned down the following statement and highlighted the C-terminus of Mcf1 in Figure 1.

“This result demonstrates that the C-terminus of Mcf1 is a potential regulatory element that prevents autoprocessing of Mcf1 prior to its translocation into the target cell.”

2.12 Page 9, paragraph 4: Since the authors could not identify clearly the location of cleavage sites 2 and 3, is it possible that this is the result of non-specific proteolysis?

We cannot exclude the possibility of non-specific proteolysis. However, we always observed a similar cleavage pattern *in vitro* and *in vivo*.

2.13 Page 9, paragraph 5: The statement that TD1 is membrane-bound during intoxication is neither supported by evidence or references and it should therefore be made clear that this is an assumption. The immediately following sentence which starts “This result demonstrates” also seems to suggest that this is something for which the authors provide evidence. There is no direct evidence for this that I can see anywhere in the manuscript, and so again the sentence should be reworded to indicate that “such a scenario would corroborate the idea...” or similar.

We modified the following sentences as suggested by the reviewer.

“In contrast to the *in vitro* conditions, where the protease effector domain can access this region and excise the putative translocation domain 1, during cell intoxication this domain is likely membrane-bound and therefore inaccessible to the protease. Such a scenario would further corroborate the idea that the conformational change during membrane penetration precedes toxin activation inside the host cells.”

2.14 I found it intriguing that the authors mentioned similarity between a region of the ABD and the BH3 domain of apoptotic proteins. This was not highlighted in the structure, and indeed was not mentioned again until page 10, paragraph 4, where the authors ultimately provide evidence that the proteolytic fragment containing this sequence has no apparent apoptotic activity. Again, in the interest of being concise, it might be better to remove mention of this from the introduction and simply state on page 10 that, despite sharing similarity with BH3-like proteins and Burkholderia lethal factor, there was no detectable cytotoxicity associated with this fragment. On this point, the authors reached the conclusion that the domain is therefore inactive, but retained for structural purposes, and while I believe it is completely justified to speculate on this point, it should be made clear that this is speculation, especially since it is based upon only a single negative result, and that the activity was tested in yeast cells, which are not the native target of Mcf.

We highlighted the BH3 domain both in the abstract and in the introduction because the previous publication (Dowling et al., Cell Microbiol 2007) suggested that this region is a major determinant of Mcf1 toxicity. We believe that it is important to provide the reader with the complete state-of-the-art information in the introduction section and therefore we prefer to keep it. However, we do agree that yeast cells are not the native target of Mcf1 and we have therefore adjusted the corresponding text accordingly:

“Therefore, we conclude that this part of Mcf1 does not induce apoptosis and is in general not an active effector, at least inside yeast cells. We speculate that this domain lost its enzymatic activity but remained an important structural element of the toxin.”

2.15 The described blurry-ness of the 2D class averages in Supp Fig.7 and Supp Fig. 8 is very hard to see at the scale that these are reproduced. A similarly comment could be made about the 2D classes they are supposed to be compared to in Supp Fig. 1. Given their importance, I would suggest showing a few, representative, enlarged 2D classes that highlight this comparison.

We have added representative enlarged 2D classes as suggested by the reviewer.

2.16 Page 13, paragraph 1: The statement that N-terminal effector domain and protease effector domain are structurally rigid. Is there evidence for this, or is this an assumption to facilitate the subsequent mechanistic modelling?

In cryo-EM, structurally rigid domains refine to higher resolution and appear at higher binarization thresholds. Indeed, as presented on supplementary figures 7c and 8c, it was true for both N-terminal effector domain and protease effector domain, suggesting their structural rigidity. To make it clearer for the reader, we introduced the following changes in the text.

“We therefore went back to further analyze our 3D classes and found that Arf binding induces a swinging movement of the activator-binding domain away from the N-terminal effector domain (Fig. 4a, Supplementary Fig. 8b, Supplementary Movie 2). Interestingly, the density for the N-terminal effector domain and the protease effector domain was of higher resolution and appeared at higher binarization thresholds, suggesting structural rigidity of these domains. Therefore, the movement of the activator-binding domain can lead to displacement of the cleavable linker towards the protease domain where it is proteolytically cleaved.”

It would also be appropriate to refer to these domains by their abbreviated names, given these were defined earlier in the manuscript.

In the first version of our manuscript, we referred to the Mcf1 domains by their abbreviation. However, many of our colleagues noticed that the abbreviated names made the article difficult to read. Therefore, considering that Nature Communications is an online journal, and our article is within recommended word count, we would prefer to use the full names of the domains.

2.17 Page 13, paragraph 2: Again, is this speculation or based on evidence? If the latter, please highlight the evidence more clearly. If it is speculation, please make it clear that this is the case.

In the first sentence of the paragraph, we make a hypothesis and highlight that it is a speculation by saying ‘we suggest’.

“On a molecular level, **we suggest** that Arf3 binding strains the helices of the activator binding domain and drives the movement of its N-terminal part.”

The following sentences describe how further movements are possible based on our structural model. To make it clearer that it is a speculation, we made the following change:

“We propose that this structural feature allows the activator-binding domain to swing away from the N-terminal effector, and to pull the cleavable linker toward the catalytic center of the protease effector domain for proteolysis (Fig. 4c and f).”

2.18 Following on, Page 13, paragraph 3: “To test this mechanism” – I believe a more appropriate terminology would be hypothesis rather than mechanism. The mutants described in the following section indeed test the hypothesis and help to establish the mechanism.

We agree with the reviewer and have changed the wording accordingly.

“To test this hypothesis, we first studied the role of the hook-loop interaction between the activator-binding and protease effector domains (Fig. 4e).”

2.19 Page 14, discussion paragraph 2: The receptor binding domains should be described as putative, since no receptor binding function has been directly demonstrated.

We agree with this remark. We have changed the text accordingly.

“However, Mcf possesses three putative receptor-binding domains instead of an extended and mobile CROPs domain, an additional activator-binding domain and a novel N-terminal effector with no homology to the glycosyltransferase of TcdA or TcdB.”

2.20 Page 15, paragraph 1: The last sentence of this paragraph makes a rather sizeable logic jump to suggest that there are major applications for Mcf1 in cancer and gene therapy, without any other related discussion or explanation of how this might be the case. The authors should provide more context around this, or ideally remove the sentence altogether as it seems to be to be an unnecessary exaggeration of significance.

We tuned down our claim by shortening the sentence.

“Thus, Mcf1 provides an example of an evolutionary scramble between different toxins of different families and highlights the potential of its modular architecture for the design of customizable delivery systems.”

2.21 Page 15, paragraph 2: The authors make the statement that their structural and functional data “together with previous analyses by other groups” all them to propose a mechanism, but again I believe the discussion of other relevant literature is a little underdone here. The sentence itself is made without citing any relevant literature and at the very least the authors should cite relevant literature that supports their proposed mechanism of action here. Indeed this entire paragraph outlining the proposed mechanism of action has no citations to any other published work, which makes it hard to appreciate the similarities to other systems, upon which the authors have presumably based their mechanism of action (at least in part). Ideally, a fuller discussion of the relevant literature would also accompany this section of the paper.

We thank the reviewer for this comment. We extended the discussion adding more information from the published articles and the corresponding citations.

2.22 Finally, and along these same lines, I was surprised not to find more discussion or comparison to other potentially related systems regarding certain aspects of the structure. To name a few:

- There is only a very limited mention of the PaTox toxin, which the authors flag as being similar to the protease effector domain.

We have added further information on the comparison with the PaTox toxin to the discussion:

“Finally, we demonstrate that the protease domain, which is normally required only for autoprocessing, is a toxic effector of Mcf1 *per se*, unlike the structurally similar CPD of the PaTox toxin from *P. asymbiotica* which was shown to increase the toxic effect of PaTox without being toxic itself.”

- The Adenylate cyclase toxins – which according to data presented in Supp Fig. 2 is a close structural homologue of (at least) two of the domains in the C-terminal region – is not mentioned anywhere in the manuscript that I could find?

We have added further information on the comparison under ‘The architecture of Mcf’ (page 6 paragraph 1)

The tail of Mcf1 ends with three β -strand-rich domains. The latter one as well as translocation domain 2 show structural similarity to the β -roll structures of RTX toxins without sharing their consensus sequence and characteristic Ca^{2+} binding⁵⁷. Since the last domain also shares structural similarity to a region of *C. difficile* toxins that has been connected to receptor binding (Supplementary Fig. 2), we called these domains putative receptor-binding domains 1-3 (RBD1-3) (Fig. 1c and d).

- The similarity to the MARTX system of *Vibrio* I believe is also slightly under-discussed, given that the authors basically posit that Mcf is a scramble between the Clostridial TcdA/B systems and the *Vibrio* MARTX system. It might be reasonable to include a comparison to MARTX alongside Supp Fig. 3, for example, given this statement?

We thank the reviewer for these suggestions. We have addressed the raised points in our discussion and included a comparison of Mcf1 and MARTX in Supplementary Fig. 3e. Furthermore, the structures are being compared in Supplementary Fig. 4a.

“Interestingly, the Mcf-like effector of MARTX from *V. vulnificus* is also activated by Arf binding. However, in that case Arf directly interacts with the protease domain (Supplementary Fig. 4a), leading to a different mechanism of activation¹⁴.”

“While the release of the N-terminal effector domain is similar to the glycosyltransferase cleavage from TcdA or TcdB^{58,59}, the further cleavages are more resembling the processing of the MARTX toxin from *V. vulnificus* that releases the Mcf-like fragment.”

Minor points

2.23 The phrase “for the first time” in the last sentence of the introduction is an unnecessary inclusion. The merits of the manuscript still stand even if this statement is omitted. Since many journals prefer that authors don’t include unqualified claims of priority, it might be better to remove this phrase.

We have removed this phrase.

2.24 Paragraph 3, page 4. There should be a comma after “Therefore” at the start of the second sentence.

Thank you. We added the comma.

2.25 Second paragraph, page 6: The last sentence should be rephrased – the words “results in Mcf1 to remain uncleaved” should simply say “remains uncleaved”.

We have modified the text as suggested.

2.26 Paragraph 2, page 7: the wording “resulted indeed” in the last sentence would be better phrased “indeed resulted”

We have modified the text as proposed.

2.27 Figure 2 is a very dense figure and it took me a long time looking at this to figure out everything that was presented here. I would encourage the authors to consider if there is a way that this information can be presented more clearly or intuitively, although I confess I don’t have any good suggestions for how to do this myself!

We have included a scheme visualizing the experiments to better connect the western blot results with the analysis in order to make it easier for the reader to read the figure.

2.28 Page 9, paragraph 4: Specify is an unusual term to use here – maybe locate or identify would be better?

Thank you. We changed “specify” to “locate”.

2.29 Page 10, paragraph 3: The last two sentences of this paragraph could be summarised much more succinctly by saying that despite sharing structural homology with ADP-ribosyltransferases, the Mcf1 NTE domain showed no detectable ADP-ribosyltransferase activity in yeast or insect cell assays. The rest of these sentences are simply saying that the control experiment worked.

We have modified the text as suggested by the reviewer.

Despite its structural homology with ADP-ribosyltransferases (Supplementary Fig. 2), the N-terminal effector domain showed no detectable ADP-ribosyltransferase activity in yeast or in

insect cells (Supplementary Fig. 5b, Fig. 3b), as well as no affinity to NAD⁺ when measured by isothermal titration calorimetry (Supplementary Fig. 6), indicating that it must have a different mechanism of action.

2.30 Page 14, paragraph 1 of the Discussion: The opening paragraph should be rephrased to make it clear that the authors are not suggesting Mcf1 is a major cause of health care-associated illnesses.

We have modified this sentence:

“In this study, we present a detailed structural and functional analysis of the Mcf1 toxin from *P. luminescens*, which allows us to confidently assign Mcf1 to the ABCD subfamily of bacterial toxins next to TcdA and TcdB toxins from *C. difficile*, which is a major cause of health care-associated diarrhea and colitis.”

Reviewers' Comments:

Reviewer #1:

Remarks to the Author:

The authors have done a nice job addresses reviewer comments and I remain enthusiastic about this work.

Reviewer #2:

Remarks to the Author:

I am satisfied that the authors have given due consideration to all of the comments raised in the first round of review and that the manuscript is appropriate for publication in its current state.